# Repeated losses of PRDM9-directed recombination despite the conservation of PRDM9 across vertebrates

Zachary Baker[1]*[†], Molly Schumer[2,3,4][†], Yuki Haba[5], Lisa Bashkirova[6], Chris Holland[4,7], Gil G Rosenthal[4,7], Molly Przeworski[1,2]*

[1]Department of Systems Biology, Columbia University, New York City, United States; [2]Department of Biological Sciences, Columbia University, New York City, United States; [3]Harvard Society of Fellows, Harvard University, Cambridge, United States; [4]Centro de Investigaciones Científicas de las Huastecas 'Aguazarca', Hidalgo, Mexico; [5]Department of Evolution, Ecology and Environmental Biology, Columbia University, New York City, United States; [6]Department of Biochemistry and Molecular Biophysics, Columbia University, New York City, United States; [7]Department of Biology, Texas A&M University, College Station, United States

*For correspondence: ztb2002@columbia.edu (ZB); mp3284@columbia.edu (MP)

[†]These authors contributed equally to this work

**Abstract** Studies of highly diverged species have revealed two mechanisms by which meiotic recombination is directed to the genome—through PRDM9 binding or by targeting promoter-like features—that lead to dramatically different evolutionary dynamics of hotspots. Here, we identify PRDM9 orthologs from genome and transcriptome data in 225 species. We find the complete PRDM9 ortholog across distantly related vertebrates but, despite this broad conservation, infer a minimum of six partial and three complete losses. Strikingly, taxa carrying the complete ortholog of PRDM9 are precisely those with rapid evolution of its predicted binding affinity, suggesting that all domains are necessary for directing recombination. Indeed, as we show, swordtail fish carrying only a partial but conserved ortholog share recombination properties with PRDM9 knock-outs.

## Introduction

Meiotic recombination is a fundamental genetic process that generates new combinations of alleles on which natural selection can act and, in most sexually-reproducing organisms, plays critical roles in the proper alignment and segregation of homologous chromosomes during meiosis (*Coop and Przeworski, 2007*; *de Massy, 2013*; *Lam and Keeney, 2014*). Meiotic recombination is initiated by a set of double strand breaks (DSBs) deliberately inflicted throughout the genome, whose repair leads to crossover and non-crossover recombination events (*Lam and Keeney, 2014*). Most of the molecular machinery involved in this process in vertebrates has been conserved since the common ancestor of plants, animals and fungi (*de Massy, 2013*). Notably, in all sexually reproducing species that have been examined, the SPO11 protein generates DSBs, which localize to histone H3 lysine K4 trimethylation marks (H3K4me3) along the genome (*Borde et al., 2009*; *Buard et al., 2009*; *Lam and Keeney, 2014*). Yet not all features of meiotic recombination are conserved across species. As one example, in many species, including all yeast, plant and vertebrate species studied to date, recombination events are localized to short intervals (of hundreds to thousands of base pairs; *Lange et al., 2016*) known as recombination hotspots, whereas in others, such as in flies or worms, the recombination landscape seems more uniform, lacking such hotspots (*Rockman and Kruglyak, 2009*; *Chan et al., 2012*; *Heil et al. 2015*)

**eLife digest** The genetic information of Eukaryotic organisms (animals, plants and fungi) is encoded on strands of DNA called chromosomes. In animals that sexually reproduce, most cells carry two copies of each chromosome, with one inherited from each of their parents. Sex cells such as sperm and egg cells are the exception, and contain only a paternal or maternal set respectively. These chromosomes are not exact copies of the parental chromosomes, but are instead combinations of both of them, generated by a process called meiotic recombination. Meiotic recombination begins by breaking the chromosomes, and the repair of those breaks shuffles DNA segments between the two chromosomes. This shuffling is known as a "recombination event".

In humans, apes and mice, the location of recombination events depends on where a protein called PRDM9 binds to the DNA. Over the course of evolution, this binding location has changed relatively rapidly so that even closely related species such as humans and chimpanzees localize recombination events to different DNA regions. In contrast, closely related species that do not produce PRDM9 tend to direct recombination events to similar DNA regions.

It remains unclear when PRDM9 first evolved its role in recombination, or why different methods of directing recombination have developed. To begin answering these questions, Baker, Schumer et al. investigated whether 225 species of vertebrates (backboned animals) have a gene that encodes PRDM9. This analysis revealed that even distantly related animals have genes that produce equivalents of the complete PRDM9 protein. However, several species have independently lost the ability to produce PRDM9. In certain other species, particular regions of the gene have been removed or shortened. Notably, only species that carry genes that contain regions called the KRAB and SSXRD domains show relatively rapid evolution of where PRDM9 binds in the DNA.

To investigate this phenomenon further, Baker, Schumer et al. constructed a map of recombination events in swordtail fish, which carry a version of the gene that lacks the KRAB and SSXRD domains. The PRDM9 protein produced by this gene does not direct where recombination events occur.

Overall, it appears that the KRAB and SSXRD domains are necessary for PRDM9 to direct meiotic recombination. Furthermore, Baker, Schumer et al. predict that those species that have complete versions of PRDM9 use this protein to localize recombination events. Knowing which species use PRDM9 in this way is the first step towards understanding why recombination mechanisms change in evolution, and with what consequences.

Among species with recombination hotspots, there are at least two mechanisms directing their location. In mammalian species, including apes, mice and likely cattle, the locations of recombination hotspots are specified by PRDM9 binding (*Baudat et al., 2010*; *Myers et al., 2010*; *Parvanov et al., 2010*; *Sandor et al., 2012*; *Great Ape Genome Project et al., 2016*). In these species, PRDM9 has four major functional domains: a KRAB, SSXRD and PR/SET domain (referred to as the SET domain in what follows), followed by a C2H2 zinc finger (ZF) array (*Figure 1*). During meiosis, PRDM9 binds sequences throughout the genome, as specified by its ZF array (reviewed in *Ségurel et al., 2011*), and the SET domain of PRDM9 makes H3K4me3 and H3K36me3 marks nearby (*Eram et al., 2014*; *Powers et al., 2016*). These actions ultimately serve to recruit SPO11 to initiate DSBs, by a mechanism that remains unknown but is associated with the presence of both histone marks (*Grey et al., 2017*; *Getun et al., 2017*) and may involve KRAB and SSXRD domains (*Parvanov et al., 2017*).

A common feature of the recombination landscape in these species is that recombination tends to be directed away from PRDM9-independent H3K4me3 peaks (*Brick et al., 2012*) and, in particular, only a small proportion of hotspots occurs at transcription start sites (TSSs; *Myers et al., 2005*; *Coop et al., 2008*). In contrast, in yeasts, plants, and vertebrate species (such as birds and canids) that lack functional PRDM9 orthologs, recombination events are concentrated at or near promoter-like features, including TSSs and CpG islands (CGIs), perhaps because they are associated with greater chromatin accessibility (*Lichten and Goldman, 1995*; *Auton et al., 2013*; *Choi et al., 2013*; *Hellsten et al., 2013*; *Lam and Keeney, 2015*; *Singhal et al., 2015*). Similarly, in mouse knockouts

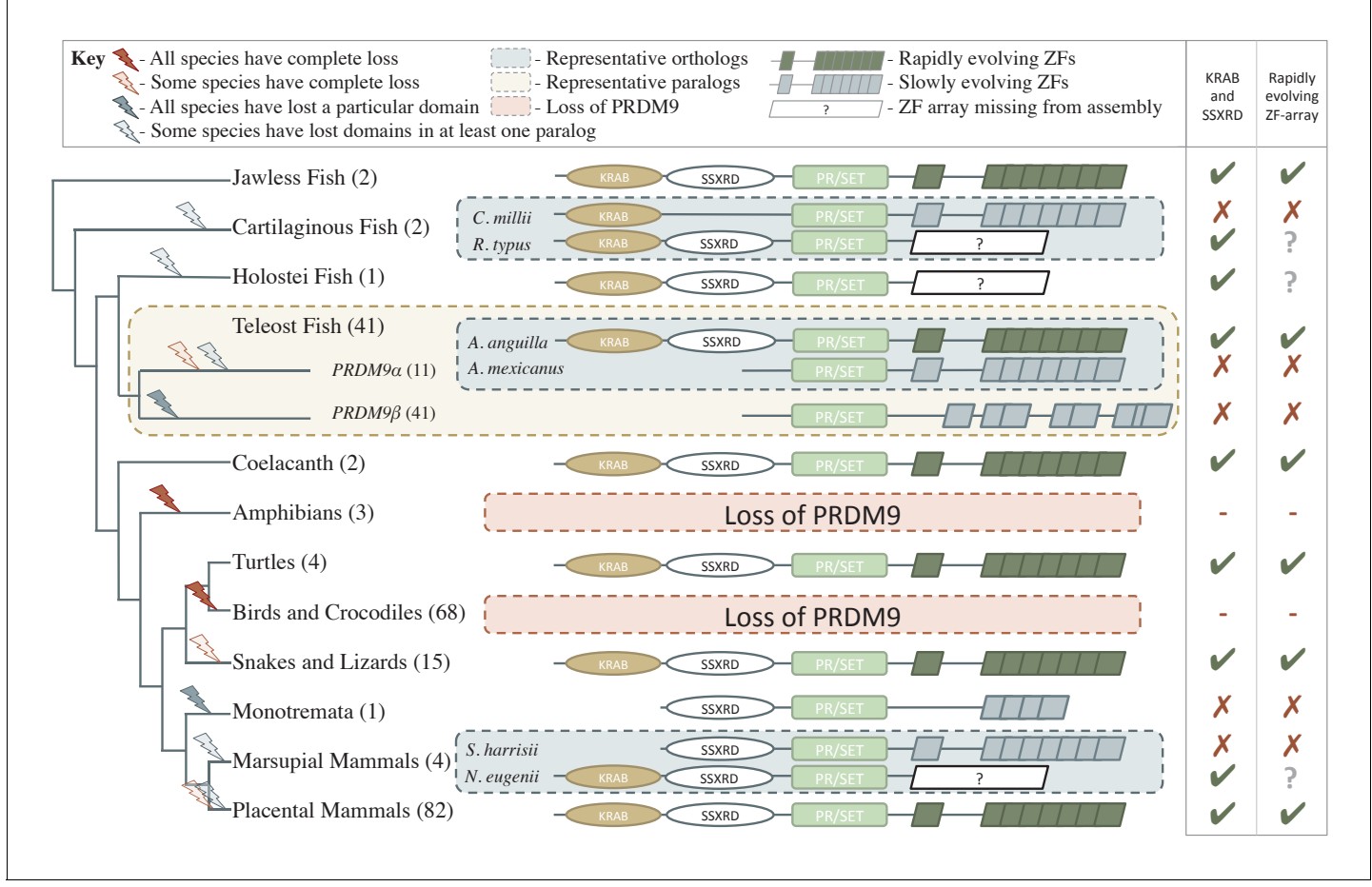

**Figure 1.** Phylogenetic distribution and evolution of PRDM9 orthologs in vertebrates. Shown are the four domains: KRAB domain (in tan), SSXRD (in white), PR/SET (in light green) and ZF (in gray/dark green; the approximate structure of identified ZFs is also shown). The number of unique species included from each taxon is shown in parenthesis. Complete losses are indicated on the phylogeny by red lightning bolts and partial losses by gray lightning bolts. Lightning bolts are shaded dark when all species in the indicated lineage have experienced the entire loss or same partial loss. Lightning bolts are shaded light when it is only true of a subset of species in the taxon. ZF arrays in dark green denote those taxa in which the ZF shows evidence of rapid evolution. White rectangles indicate cases where we could not determine whether the ZF was present, because of the genome assembly quality. For select taxa, we present the most complete PRDM9 gene found in two examplar species. Within teleost fish, we additionally show a PRDM9 paralog that likely arose before the common ancestor of this taxon; in this case, the number of species observed to have each paralog is in paranthesis. Although the monotremata ZF is shaded gray, it was not included in our analysis of rapid evolution because of its small number of ZFs.

The following figure supplements are available for figure 1:

**Figure supplement 1.** Phylogenetic approach to identifying PRDM9 orthologs and related gene families.

**Figure supplement 2.** Neighbor-joining (NJ) guide tree based on the SET domain.

**Figure supplement 3.** Expression levels of genes with a known role in meiotic recombination in testes of three examplar species: human, swordtail fish and bearded dragon (a lizard).

**Figure supplement 4.** Amino acid diversity as a function of amino acid position in the ZF alignment for six examplar species.

**Figure supplement 5.** Examples of differences in computationally predicted PRDM9 binding motifs for species from three taxa.

for PRDM9, recombination events appear to default to promoter-like features that carry H3K4me3 peaks (*Brick et al., 2012*; *Narasimhan et al., 2016*).

The two mechanisms by which recombination events are targeted to the genome are associated with dramatic differences in the evolution of recombination hotspots. When recombination is directed by PRDM9, hotspot locations are not shared between closely related ape species or between mouse subspecies and differ even among human populations (*Ptak et al., 2004*; *Myers et al., 2005*; *Ptak et al., 2005*; *Coop et al., 2008*; *Hinch et al., 2011*; *Auton et al., 2012*; *Great Ape Genome Project et al., 2016*). This rapid evolution appears to be driven by two phenomena. First, the binding specificity of the PRDM9 ZF leads to the existence of 'hotter' and 'colder' alleles, that is, sequences that are more or less likely to be bound by PRDM9 (*Myers et al., 2008*). In heterozygotes carrying a colder and a hotter allele, this asymmetry in binding leads to the hotter alleles more often experiencing a DSB (*Baker et al., 2015*; *Davies et al., 2016*). Since repair mechanisms use the intact, colder allele as a template, the sequences to which PRDM9 binds are preferentially lost (*Boulton et al., 1997*; *Kauppi et al., 2005*). This process of under-transmission of the hotter allele in hot/cold heterozygotes acts analogously to selection for the colder allele (*Nagylaki and Petes, 1982*) and is thus expected to drive the rapid loss of hotspots from the population (leading to the 'hotspot paradox'; *Pineda-Krch and Redfield, 2005*; *Coop and Myers, 2007*), consistent with empirical observations in humans and mice (*Berg et al., 2010*; *Myers et al., 2010*; *Baker et al., 2015*; *Smagulova et al., 2016*).

In addition to this loss of hotspots in *cis*, changes in the PRDM9 binding domain can also lead to the rapid loss—and gain—of whole sets of hotspots. Interestingly, PRDM9 has the fastest evolving C2H2 ZF array in the mouse and human genomes (*Oliver et al., 2009*; *Myers et al., 2010*). More generally, mammalian PRDM9 genes show strong evidence of positive selection at known DNA-binding sites of ZFs (*Oliver et al., 2009*). Thus, in mammals carrying PRDM9, individual hotspots are lost quickly over evolutionary time, but changes in the PRDM9 ZF generate novel sets of hotspots, leading to rapid turnover in the fine-scale recombination landscape between populations and species.

The mechanism driving the rapid evolution of the PRDM9 ZF is unclear. One hypothesis is that the under-transmission of hotter alleles eventually leads to the erosion of a sufficient number of hotspots that the proper alignment or segregation of homologs during meiosis is jeopardized, strongly favoring new ZF alleles (*Coop and Myers, 2007*; *Myers et al., 2010*; *Ubeda and Wilkins, 2011*). Whether hotspot loss would exert a sufficiently strong and immediate selection pressure to explain the very rapid evolution of the PRDM9 ZF remains unclear. An alternative explanation has emerged recently from the finding that in mice, widespread asymmetric binding by PRDM9 on the two homologs is associated with hybrid sterility (*Davies et al., 2016*; *Smagulova et al., 2016*). Since older PRDM9 motifs are more likely to have experienced erosion and hence to be found in heterozygotes for hotter and colder alleles, there may be an immediate advantage to new ZFs that lead to greater symmetry in PRDM9 binding (*Davies et al., 2016*). Regardless of the explanation, the rapid evolution of the PRDM9 ZF is likely tied to its role in recombination.

Conversely, in species that do not use PRDM9 to direct meiotic recombination events, the rapid evolution of recombination hotspots is not seen. In birds that lack an ortholog of PRDM9, the locations of recombination hotspots are conserved over long evolutionary time scales. Similarly, both the location and heats of recombination hotspots are conserved across highly diverged yeast species, in which H3K4me3 marks are made by a single gene without a DNA binding domain (*Lam and Keeney, 2015*). In these taxa, it remains unknown whether the coincidence of recombination with functional genomic elements, such as TSSs and CGIs, is facilitated by specific binding motifs or simply by greater accessibility of the recombination machinery to these genomic regions (*Brick et al., 2012*; *Auton et al., 2013*; *Choi et al., 2013*; *Lam and Keeney, 2015*; *Singhal et al., 2015*). Even if there are specific motifs that increase rates of recombination near functional genomic elements, they are likely to have important, pleiotropic consequences (*Nicolas et al., 1989*). Thus, there may be a strong countervailing force to the loss of hotspots by under-transmission of hotter alleles in these cases, leading to the evolutionary stability of hotspots.

These observations sketch the outline of a general pattern, whereby species that do not use PRDM9 to direct recombination target promoter-like features and have stable fine-scale recombination landscapes, whereas those that employ PRDM9 tend to recombine away from promoters and experience rapid turnover of hotspot locations. This dramatic difference in the localization of

hotspots and their evolutionary dynamics has important evolutionary consequences for genome structure and base composition, for linkage disequilibrium (LD) levels along the genome, as well as for introgression patterns in naturally occurring hybrids (*Fullerton et al., 2001*; *McVean et al., 2004*; *Duret and Galtier, 2009*; *Janoušek et al., 2015*). It is therefore important to establish the generality of these two mechanisms and characterize their distribution across species.

To date, studies of fine-scale recombination rates are limited to a handful of organisms. In particular, although it has been previously reported that the PRDM9 gene arose early in metazoan evolution (*Oliver et al., 2009*), direct evidence of its role in recombination is limited to placental mammals (mice, primates and more circumstantially cattle). It remains unknown which species carry an intact ortholog and, more broadly, when PRDM9-directed recombination is likely to have arisen. To address these questions, we investigated the PRDM9 status of 225 species of vertebrates, using a combination of genome sequences and RNAseq data.

## Results

### Initial identification of PRDM9 orthologs in vertebrates

In order to identify which species have PRDM9 orthologs, we searched publically available nucleotide and whole genome sequences to create a curated dataset of vertebrate PRDM9 sequences. To this end, we implemented a *blastp*-based approach against the *RefSeq* database, using human PRDM9 as a query sequence (see Materials and methods for details). We supplemented this dataset with 44 genes strategically identified from 30 whole genome assemblies and seven genes identified from de novo assembled transcriptomes from testis of five species lacking genome assemblies (see Materials and methods for details). Neighbor joining (NJ) and maximum likelihood trees were built using identified SET domains to distinguish *bona fide* PRDM9 orthologs from members of paralagous gene families and to characterize the distribution of PRDM9 duplication events (*Figure 1—figure supplement 1* and *2*). Since the placement of the major taxa used in our analysis is not controversial, in tracing the evolution of PRDM9 orthologs, we assumed that the true phylogenetic relationships between taxa are those reported by several recent papers (synthesized by the TimeTree project; *Hedges et al., 2015*).

This approach identified 227 PRDM9 orthologs (*Supplementary file 1A,B*), found in jawless fish, cartilaginous fish, bony fish, coelacanths, turtles, snakes, lizards, and mammals. We confirmed the absence of PRDM9 in all sampled birds and crocodiles (*Oliver et al., 2009*; *Singhal et al., 2015*), the absence of non-pseudogene copies in canids (*Oliver et al., 2009*; *Muñoz-Fuentes et al., 2011*), and additionally were unable to identify PRDM9 genes in amphibians (*Figure 1*), despite targeted searches of whole genome sequences (*Supplementary file 1B*).

We further inferred an ancient duplication of PRDM9 in the common ancestor of teleost fish, apparently coincident with the whole genome duplication that occurred in this group (*Figure 1*, *Figure 2*). We used both phylogenetic methods and analysis of the ZF structure to distinguish these copies (see *Figure 2—figure supplement 1*, Materials and methods) and refer to them as PRDM9$\alpha$ and PRDM9$\beta$ in what follows. While PRDM9$\beta$ orthologs were identified in each species of teleost fish examined, we were unable to identify PRDM9$\alpha$ type orthologs within three major teleost taxa, suggesting at minimum three losses of PRDM9$\alpha$ type orthologs within teleost fish (*Figure 2*, *Supplementary file 1A*). Several additional duplication events appear to have occurred more recently in other vertebrate groups, including in jawless fish, cartilaginous fish, bony fish, and mammals (*Supplementary file 1A*).

### Expression of PRDM9 in the germline of major vertebrate groups

Since a necessary condition for PRDM9 to play a role in meiotic recombination is for it to be expressed in the germline, we looked for PRDM9 in expression data from testis tissues in order to confirm its presence. We focused on testis expression rather than ovaries because although both obviously contain germline cells, preliminary analyses suggested that meiotic gene expression is more reliably detected in testes (see Materials and methods). We selected 23 representative species, spanning each major vertebrate group, with publically available testis expression or testis RNA-seq (*Supplementary file 2A*); we also generated testis RNA-seq data for two species of bony fish (see Materials and methods). In teleost fish with both PRDM9$\alpha$ and PRDM9$\beta$ genes, we were able to

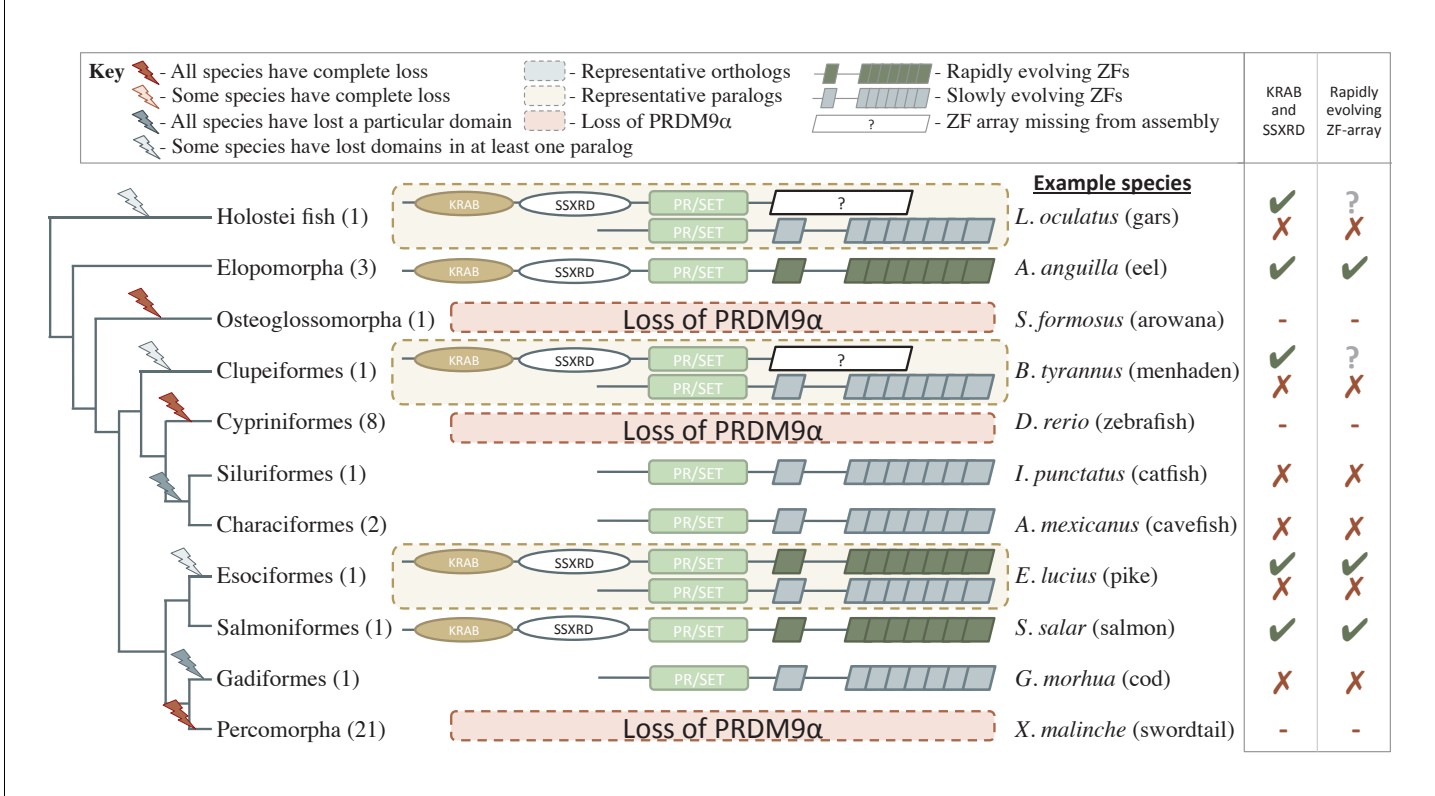

**Figure 2.** Phylogenetic distribution and functional domains of PRDM9α orthologs in teleost fish and in holostean fish that are outgroups to the PRDM9α/PRDM9β duplication event. Shown are the four domains: KRAB domain (in tan), SSXRD (in white), PR/SET (in light green) and ZF (in gray/dark green; the approximate structure of identified ZFs is also shown). The number of unique species included from each taxon is shown in parenthesis. Complete losses are indicated on the phylogeny by red lightning bolts and partial losses by gray lightning bolts. Lightning bolts are shaded dark when all species in the indicated lineage have experienced the loss. Lightning bolts are shaded light when it is only true of a subset of species in the taxon. ZF arrays in dark green denote those taxa in which the ZF shows evidence of rapid evolution. White rectangles indicate cases where we could not determine whether the ZF was present, because of the genome assembly quality. While many taxa shown have more than one PRDM9α ortholog, the genes identified from each species generally have similar domain architectures. Exceptions include Clupeiformes, Esociformes, and Holostean fish, for which two alternative forms of PRDM9α paralogs are shown. Based on this distribution, we infer that the common ancestor of ray-finned fish likely had a rapidly evolving and complete PRDM9α ortholog.

The following figure supplements are available for figure 2:

**Figure supplement 1.** Section of maximum-likelihood phylogeny of the SET domain showing bony fish PRDM9 orthologs α and β.

**Figure supplement 2.** Analysis of ZF evolution in PRDM9β.

detect either the expression of both orthologs or only expression of PRDM9α orthologs. In species of teleost fish with only PRDM9β genes, we consistently identified expression of PRDM9β genes. More generally, we were able to identify PRDM9 expression in nearly all RNA-seq datasets from species in which the genome carried a putative ortholog, the elephant shark (*Callorhinchus milii*) being the sole exception (**Supplementary file 2B,C**).

## Confirmation of PRDM9 loss events

Concerned that absences of PRDM9 observed in some species could reflect lower quality genome assemblies rather than true loss events, we also used testis RNAseq data to investigate putative losses of PRDM9 in amphibians and fish (PRDM9α). To this end, we relied on the fact that when PRDM9 is present, it is detectable in RNAseq data from the whole testis of vertebrates (see above). Our approach was to analyze testis transcriptome data from species lacking PRDM9 sequences in their genome assemblies, using an analysis that is not biased by the genome assembly (see Materials

and methods). For each species, we confirmed that the dataset captured the appropriate cell populations and provided sufficient power to detect transcripts that are expressed during meiosis at levels comparable to PRDM9 in mammals (*Figure 1—figure supplement 3*, *Supplementary file 2B,D*). With this approach, we were able to find support for the loss of PRDM9 in salamanders (*Cynops pyrrhogaster, Ambystoma mexicanum*) and frogs (*Xenopus tropicalis*). Because of the paucity of amphibian genomes, however, it is not clear whether or not these examples represent a widespread loss of PRDM9 within amphibians or more recent, independent losses. Within bony fish, we were able to confirm the three independent losses of PRDM9α type orthologs in one species each of percomorph (*Xiphophorus birchmanni*), cypriniform (*Danio rerio*) and osteoglossomorph fish (*Osteoglossum bicirrhosum*). Thus, in all cases with sufficient power to detect expression of PRDM9 in testes data, our findings were consistent with inferences based on genome sequence data.

## Inferences of PRDM9 domain architecture

PRDM9 orthologs identified in jawless fish, some bony fish, coelacanths, lizards, snakes, turtles, and placental mammals have a complete domain structure, consisting of KRAB, SSXRD and SET domains, as well as a C2H2 ZF array. The phylogenetic relationships between these species suggest that a complete PRDM9 ortholog was present in the common ancestor of vertebrates (*Figure 1*).

Despite its widespread taxonomic distribution, however, the complete domain structure was not found in several of the 149 sampled lineages with PRDM9 orthologs (*Figure 1*; in addition to the complete losses of the gene described above). Instances include the absence of the SSXRD domain in some cartilaginous fish (see Materials and methods), absence of both KRAB and SSXRD domains in PRDM9β orthologs (*Figure 1*) and in PRDM9α orthologs found distributed throughout the teleost fish phylogeny (*Figure 2*, *Figure 2—figure supplement 1*), and the absence of the KRAB domain in monotremata (*Ornithorhynchus anatinus*) and marsupial mammals (*Sarcophilus harrisii*, *Figure 1*; *Supplementary file 1A*).

Because these frequent N-terminal losses could be the result of assembly or gene prediction errors, we sought to confirm them by systematically searching genomes and transcriptomes for evidence of these missing domains (see Materials and methods). We required not only that missing domains homologous to PRDM9 be absent from the genome in a whole genome search, but also that the missing domain not be present in the transcriptome, when other domains of PRDM9 were. This approach necessarily limits our ability to verify putative losses when there are no suitable transcriptome data, but nonetheless allowed us to confirm the losses of the KRAB and SSXRD domains in a PRDM9 ortholog from holostean fish (*Lepisosteus oculatus*), in all PRDM9β orthologs from teleost fish (*Figure 1*), in PRDM9α orthologs that lost their complete domain structure in several taxa of teleost fish (*Gadus morhua*, *Astyanax mexicanus*, *Ictalurus punctatus*, *Esox lucius*; *Supplementary file 2C*), as well as losses of the KRAB domain in two PRDM9 orthologs identified in monotremata (both in *O. anatinus*, *Supplementary file 2C*), indicating a minimum of six N-terminal domain losses within vertebrates.

For representative cases where we were able to confirm missing N-terminal domains, we further investigated whether the truncated genes had become pseudogenes by testing whether the ratio of nonsynonymous to synonymous substitutions in the SET domain is significantly different than 1 (see Materials and methods). In all cases of N-terminal truncation, the partial PRDM9 shows evidence of functional constraint (i.e., dN/dS <1, where dN is the rate of amino-acid substitutions and dS of synonymous substitutions; see Materials and methods for more details). This conservation is most strikingly seen in teleost fish, in which a partial PRDM9 ortholog has been evolving under constraint for hundreds of millions of years (*Figure 1*, *Figure 2—figure supplement 1*, *Supplementary file 3A*). These observations suggest that in these species, PRDM9 has an important function that it performs without KRAB or SSXRD domains. Moreover, these cases provide complementary observations to full PRDM9 knockouts in amphibians and archosaurs, allowing the roles of specific domains to be dissected.

## Evidence for rapid evolution of PRDM9 binding specificity

Rapid evolution of the PRDM9 ZF array has been reported previously in all species with evidence for PRDM9-directed recombination, including cattle, apes and mice. While it is not known whether this rapid evolution is a necessary consequence of its role in recombination, plausible models suggest it

is likely to be (see Introduction). If so, we expect species with PRDM9-directed recombination to show evidence for rapidly-evolving PRDM9 ZF arrays and can use this feature to hone in on the subset of PRDM9 orthologs most likely to play a role in recombination.

To this end, we characterized the rapid evolution of the PRDM9 ZF in terms of the proportion of amino acid diversity within the ZF array that occurs at DNA-binding sites (using a modification of the approach proposed by *Oliver et al., 2009*). This summary statistic is sensitive to both rapid amino acid evolution at DNA binding sites and concerted evolution between the individual ZFs (see Materials and methods). Using this statistic, placental mammals that have PRDM9-directed recombination show exceptionally high rates of evolution of the PRDM9 ZF compared to other ZFs (*Table 1*; *Baudat et al., 2010*; *Myers et al., 2010*; *Parvanov et al., 2010*). Moreover, two of six cattle PRDM9 orthologs that we identified were previously associated with interspecific variation in recombination phenotypes (*Supplementary file 3B*; *Sandor et al., 2012*; *Ma et al., 2015*), and both are seen to be rapidly evolving (*Table 1*, *Supplementary file 3B*).

In addition to placental mammals, PRDM9 orthologs in jawless fish, some bony fish (Salmoniformes, Esociformes, Elopomorpha), turtles, snakes, lizards, and coelacanths show similarly elevated values of this statistic (*Figure 1—figure supplement 4*). In fact, PRDM9 is the most rapidly evolving ZF gene genome-wide in most species in these taxa and all PRDM9 orthologs with the complete domain structure were in the top 5% of the most rapidly evolving ZFs in their respective genomes (*Table 1*, *Supplementary file 3B*). In contrast, evidence of such rapid evolution is absent from other taxa of bony fish, including all PRDM9$\beta$ orthologs and partial PRDM9$\alpha$ orthologs, as well as from the putatively partial PRDM9 orthologs found in the elephant shark, the Tasmanian devil, and in several species of placental mammals (see Materials and methods for details). We only observed one instance (little brown bat, *Myotis lucifugus*) in which a partial PRDM9 ortholog was evolving unusually rapidly (*Table 1*); in this case, we were unable to confirm the loss of the missing KRAB domain (see Materials and methods), so it remains possible this ortholog is in fact intact. In summary, with one possible exception, species show evidence of rapid evolution of the ZF binding affinity if and only if they carry the intact PRDM9 ortholog found in placental mammals. This concordance of rapid evolution with the complete domain structure is highly unlikely by chance (taking into account the phylogenetic relationship between orthologs, p<10$^{-6}$; see Materials and methods). Assuming that rapid evolution of the ZF is indicative of PRDM9-directed recombination, these observations carry two implications: KRAB and SSXRD domains are required for this role and non-mammalian species such as turtles or snakes also use PRDM9 to direct recombination.

## Analysis of SET domain catalytic residues

While partial orthologs of PRDM9 have lost one or both of their N-terminal domains, they retain the SET and ZF domains known to play a role in recombination, are under purifying selection, and are expressed in testis. In principle then, these partial orthologs could still play a role in directing recombination. To evaluate this possibility, we started by examining whether the catalytic activities of the SET domains of partial PRDM9 orthologs are conserved. We did so because the catalytic specificities of PRDM9 are believed to be important to its role in directing recombination: two marks made by the SET domain of PRDM9, H3K4me3 and H3K36me3, are associated with hotspot activity in mammals (*Powers et al., 2016*; *Grey et al., 2017*; *Yamada et al., 2017*; *Getun et al., 2017*) and the human PRDM9 is unusual in being able to add methyl groups to different lysine residues of the same nucleosomes, when most other methyltransferase genes are responsible for only a single mark (*Eram et al., 2014*; *Powers et al., 2016*).

Specifically, we focused on three tyrosine residues shown to be important for the catalytic specificities of the human PRDM9 gene (Y276, Y341 and Y357; see Materials and methods and *Supplementary file 1A*; *Wu et al., 2013*) and asked if those residues were conserved across vertebrates. Loss of individual residues is not necessarily evidence for loss of catalytic activity, as compensatory changes may have occurred. For example, a substitution at Y357 of PRDM7 has led to the loss of H3K36me3 specificity, but H3K4me3 activity appears to have been retained through compensatory substitutions (*Blazer et al., 2016*). Nonetheless, PRDM9 orthologs with substitutions at these residues are unlikely to utilize the same catalytic mechanisms as human PRDM9 for any methyltransferase activity that they retain.

We found that each of the three residues is broadly conserved across the vertebrate phylogeny, with substitutions observed in only 57 of 227 PRDM9 orthologs, including 11 genes from placental

**Table 1.** Evolution of the ZF in PRDM9 orthologs with different domain architectures. PRDM9 orthologs for which an empirical comparison dataset is available are ordered by their domain structures: from the top, we present cases of complete PRDM9 orthologs with KRAB-SSXRD-SET domains; partial orthologs putatively lacking KRAB or SSXRD domains or partial orthologs lacking both; then those containing only the SET domain. A row is shaded green if the ZF is in the top 5% most rapidly evolving C2H2 ZF in the species, as summarized by the proportion of amino-acid diversity at DNA-binding sites, and is blue if it is ranked first. A complete PRDM9 ortholog from dolphins (*Balaenoptera acuforostrata scammoni*) is shaded in gray because there is no amino acid diversity between ZFs of the tandem array. The empirical rank is also shown, as are the number of PRDM9 orthologs identified in the species. Asterisks indicate PRDM9 orthologs known to play a role in directing recombination. For PRDM9 genes from teleost fish, under major group, we additionally indicate whether or not the gene is a PRDM9α or PRDM9β gene.

| Organism | Major group | PRDM9 structure | Proportion AA diversity at DNA-binding sites | Rank | Number of PRDM9 genes from species | Number of ZF genes evaluated from species |
|---|---|---|---|---|---|---|
| *Balaenoptera acutorostrata scammoni* | placental | KRAB-SSXRD-SET | NA | NA | 1 | 272 |
| *Bison bison bison* | placental | KRAB-SSXRD-SET | 0.667 | 1 | 1 | 285 |
| *Bos taurus** (chr1) | placental | KRAB-SSXRD-SET | 0.684 | 1 | 3 | 313 |
| *Bos taurus* (chrX) | placental | KRAB-SSXRD-SET | 0.414 | 6 | 3 | 313 |
| *Bos taurus** (chrX) | placental | KRAB-SSXRD-SET | 0.414 | 7 | 3 | 313 |
| *Bubalus bubalis* | placental | KRAB-SSXRD-SET | 0.667 | 1 | 1 | 268 |
| *Chelonia mydas* | turtle | KRAB-SSXRD-SET | 0.414 | 11 | 1 | 235 |
| *Chlorocebus sabaeus* | placental | KRAB-SSXRD-SET | 0.500 | 1 | 1 | 344 |
| *Chrysemys picta bellii* | turtle | KRAB-SSXRD-SET | 0.478 | 1 | 1 | 308 |
| *Cricetulus griseus* | placental | KRAB-SSXRD-SET | 0.781 | 3 | 1 | 259 |
| *Dasypus novemcinctus* | placental | KRAB-SSXRD-SET | 0.614 | 1 | 1 | 289 |
| *Dipodomys ordii* | placental | KRAB-SSXRD-SET | 0.567 | 1 | 1 | 194 |
| *Esox lucius* | teleost fish (α) | KRAB-SSXRD-SET | 0.455 | 1 | 4 | 234 |
| *Fukomys damarensis* | placental | KRAB-SSXRD-SET | 0.430 | 3 | 1 | 227 |
| *Homo sapiens** | placental | KRAB-SSXRD-SET | 0.687 | 1 | 1 | 357 |
| *Latimeria chalumnae* | coelacanth | KRAB-SSXRD-SET | 0.545 | 2 | 1 | 227 |
| *Loxodonta africana* | placental | KRAB-SSXRD-SET | 0.617 | 1 | 1 | 381 |
| *Macaca fascicularis* | placental | KRAB-SSXRD-SET | 0.680 | 1 | 1 | 364 |
| *Macaca mulatta* | placental | KRAB-SSXRD-SET | 0.645 | 1 | 1 | 366 |
| *Marmota marmota marmota* | placental | KRAB-SSXRD-SET | 0.483 | 1 | 1 | 277 |
| *Microcebus murinus* | placental | KRAB-SSXRD-SET | 1.000 | 1 | 1 | 326 |
| *Mus musculus** | placental | KRAB-SSXRD-SET | 0.910 | 1 | 1 | 224 |
| *Nannospalax galili* | placental | KRAB-SSXRD-SET | 1.000 | 1 | 1 | 307 |
| *Octodon degus* | placental | KRAB-SSXRD-SET | 0.333 | 5 | 3 | 227 |
| *Octodon degus* | placental | KRAB-SSXRD-SET | 0.331 | 6 | 3 | 227 |
| *Ovis aries* | placental | KRAB-SSXRD-SET | 0.615 | 1 | 2 | 252 |
| *Ovis aries* | placental | KRAB-SSXRD-SET | 0.398 | 4 | 2 | 252 |
| *Ovis aries musimon* | placental | KRAB-SSXRD-SET | 0.353 | 12 | 1 | 285 |
| *Papio anubis* | placental | KRAB-SSXRD-SET | 0.585 | 1 | 1 | 404 |
| *Pelodiscus sinensis* | turtle | KRAB-SSXRD-SET | 0.692 | 1 | 1 | 221 |
| *Peromyscus maniculatus bairdii* | placental | KRAB-SSXRD-SET | 1.000 | 1 | 1 | 243 |
| *Protobothrops mucrosquamatus* | squamata | KRAB-SSXRD-SET | 0.462 | 5 | 1 | 195 |
| *Python bivittatus* | squamata | KRAB-SSXRD-SET | 0.571 | 1 | 1 | 206 |
| *Rattus norvegicus* | placental | KRAB-SSXRD-SET | 0.570 | 1 | 1 | 255 |
| *Rousettus aegyptiacus* | placental | KRAB-SSXRD-SET | 0.742 | 1 | 1 | 258 |
| *Salmo salar* | teleost fish (α) | KRAB-SSXRD-SET | 0.538 | 9 | 4 | 510 |

*Table 1 continued on next page*

*Table 1 continued*

| Organism | Major group | PRDM9 structure | Proportion AA diversity at DNA-binding sites | Rank | Number of PRDM9 genes from species | Number of ZF genes evaluated from species |
|---|---|---|---|---|---|---|
| *Salmo salar* | teleost fish (α) | KRAB-SSXRD-SET | 0.500 | 11 | 4 | 510 |
| *Sus scrofa* | placental | KRAB-SSXRD-SET | 0.542 | 1 | 1 | 248 |
| *Thamnophis sirtalis* | squamata | KRAB-SSXRD-SET | 0.459 | 3 | 1 | 179 |
| *Tupaia chinensis* | placental | KRAB-SSXRD-SET | 1.000 | 1 | 1 | 249 |
| *Tursiops truncatus* | placental | KRAB-SSXRD-SET | 0.939 | 1 | 1 | 233 |
| *Myotis lucifugus* | placental | SSXRD-SET | 0.524 | 1 | 2 | 308 |
| *Myotis lucifugus* | placental | SSXRD-SET | 0.310 | 68 | 2 | 308 |
| *Octodon degus* | placental | SSXRD-SET | 0.282 | 46 | 3 | 227 |
| *Sarcophilus harrisii* | marsupial | SSXRD-SET | 0.224 | 277 | 2 | 344 |
| *Callorhinchus millii* | cartilaginous fish | KRAB-SET | 0.314 | 6 | 1 | 63 |
| *Astyanax mexicanus* | teleost fish (α) | SET | 0.258 | 60 | 2 | 158 |
| *Astyanax mexicanus* | teleost fish (β) | SET | 0.167 | 152 | 2 | 158 |
| *Clupea harengus* | teleost fish (α) | SET | 0.279 | 6 | 4 | 118 |
| *Clupea harengus* | teleost fish (α) | SET | 0.278 | 7 | 4 | 118 |
| *Clupea harengus* | teleost fish (α) | SET | 0.274 | 10 | 4 | 118 |
| *Clupea harengus* | teleost fish (β) | SET | 0.158 | 114 | 4 | 118 |
| *Cynoglossus semilaevis* | teleost fish (β) | SET | 0.182 | 80 | 1 | 107 |
| *Danio rerio* | teleost fish (β) | SET | 0.179 | 345 | 1 | 367 |
| *Esox lucius* | teleost fish (α) | SET | 0.295 | 32 | 4 | 234 |
| *Esox lucius* | teleost fish (β) | SET | 0.192 | 176 | 4 | 234 |
| *Esox lucius* | teleost fish (β) | SET | 0.192 | 177 | 4 | 234 |
| *Fundulus heteroclitus* | teleost fish (β) | SET | 0.189 | 158 | 1 | 206 |
| *Haplochromis burtoni* | teleost fish (β) | SET | 0.180 | 148 | 1 | 168 |
| *Ictalurus punctatus* | teleost fish (α) | SET | 0.320 | 14 | 8 | 140 |
| *Ictalurus punctatus* | teleost fish (α) | SET | 0.319 | 15 | 8 | 140 |
| *Ictalurus punctatus* | teleost fish (α) | SET | 0.306 | 24 | 8 | 140 |
| *Ictalurus punctatus* | teleost fish (α) | SET | 0.303 | 25 | 8 | 140 |
| *Ictalurus punctatus* | teleost fish (α) | SET | 0.286 | 33 | 8 | 140 |
| *Ictalurus punctatus* | teleost fish (α) | SET | 0.276 | 39 | 8 | 140 |
| *Ictalurus punctatus* | teleost fish (α) | SET | 0.253 | 55 | 8 | 140 |
| *Ictalurus punctatus* | teleost fish (β) | SET | 0.179 | 127 | 8 | 140 |
| *Larimichthys crocea* | teleost fish (β) | SET | 0.192 | 70 | 1 | 115 |
| *Lepisosteus oculatus* | holostei fish | SET | 0.223 | 48 | 1 | 106 |
| *Maylandia zebra* | teleost fish (β) | SET | 0.173 | 161 | 1 | 176 |
| *Neolamprologus brichardi* | teleost fish (β) | SET | 0.173 | 141 | 1 | 152 |
| *Nothobranchius furzeri* | teleost fish (β) | SET | 0.180 | 245 | 1 | 266 |
| *Notothenia coriiceps* | teleost fish (β) | SET | 0.167 | 83 | 1 | 87 |
| *Oreochromis niloticus* | teleost fish (β) | SET | 0.173 | 173 | 1 | 190 |
| *Oryzias latipes* | teleost fish (β) | SET | 0.213 | 104 | 1 | 191 |
| *Otolemur garnettii* | placental | SET | 0.266 | 121 | 1 | 285 |
| *Poecilia formosa* | teleost fish (β) | SET | 0.191 | 184 | 1 | 242 |
| *Poecilia latipinna* | teleost fish (β) | SET | 0.191 | 175 | 1 | 235 |
| *Poecilia mexicana* | teleost fish (β) | SET | 0.191 | 187 | 1 | 244 |

*Table 1 continued on next page*

*Table 1 continued*

| Organism | Major group | PRDM9 structure | Proportion AA diversity at DNA-binding sites | Rank | Number of PRDM9 genes from species | Number of ZF genes evaluated from species |
|---|---|---|---|---|---|---|
| *Poecilia reticulata* | teleost fish (β) | SET | 0.191 | 162 | 1 | 212 |
| *Pundamilia nyererei* | teleost fish (β) | SET | 0.173 | 134 | 1 | 147 |
| *Pygocentrus nattereri* | teleost fish (α) | SET | 0.331 | 12 | 2 | 142 |
| *Pygocentrus nattereri* | teleost fish (β) | SET | 0.179 | 124 | 2 | 142 |
| *Salmo salar* | teleost fish (β) | SET | 0.188 | 411 | 4 | 510 |
| *Salmo salar* | teleost fish (β) | SET | 0.180 | 454 | 4 | 510 |
| *Sinocyclocheilus anshuiensis* | teleost fish (β) | SET | 0.185 | 224 | 2 | 284 |
| *Sinocyclocheilus anshuiensis* | teleost fish (β) | SET | 0.185 | 225 | 2 | 284 |
| *Sinocyclocheilus grahami* | teleost fish (β) | SET | 0.185 | 211 | 1 | 271 |
| *Sinocyclocheilus rhinocerous* | teleost fish (β) | SET | 0.185 | 208 | 2 | 269 |
| *Sinocyclocheilus rhinocerous* | teleost fish (β) | SET | 0.185 | 209 | 2 | 269 |
| *Takifugu rubripes* | teleost fish (β) | SET | 0.188 | 66 | 1 | 98 |
| *Xiphophorus maculatus* | teleost fish (β) | SET | 0.191 | 117 | 1 | 158 |

mammals and 46 genes from bony fish. Strikingly, however, none of these substitutions occur in a complete PRDM9 ortholog containing KRAB, SSXRD, SET and ZF domains. Within mammals, the majority of PRDM9 orthologs that experienced these substitutions lack the ZF array entirely, including eight PRDM7 genes from primates, which share a substitution at Y357, and one PRDM9 ortholog from a bat (*Miniopterus natalensis*) that carries a substitution at Y276. Others are lacking the KRAB domain, including PRDM9 orthologs identified from a lemur (*Galeopterus variegatus*) and a rodent (*Octodon degus*), which carry substitutions at Y276 and Y357, respectively.

Within bony fish, we identified 46 PRDM9 orthologs with substitutions at one or more of these residues, including the partial PRDM9 ortholog from holosteans (see above) and all PRDM9β orthologs in teleosts (*Supplementary file 1A*). The distribution of substitutions at these residues within PRDM9β genes suggests that numerous independent substitution events have occurred in this gene family following the loss of KRAB and SSXRD domains (*Figure 3*). In contrast, no substitutions were observed at these residues in any PRDM9α ortholog, regardless of their domain architecture. These observations could be consistent with a lack of constraint on the ancestral methyltransferase activities of PRDM9 in PRDM9β genes after the PRDM9α/PRDM9β duplication event (or conceivably an indication that there has been convergent evolution towards a new functional role). Thus, PRDM9β genes not only lack KRAB and SSXRD domains, they likely lack some methyltransferase activity of the SET domain.

## Fish species with a partial PRDM9 ortholog share broad patterns of recombination with species that lack PRDM9

To more directly test the hypothesis that the partial ortholog of PRDM9 does not direct recombination, we examined patterns of crossing-over in naturally-occurring swordtail fish hybrids (*X. birchmanni* x *X. malinche*; see Materials and methods). Like other percomorphs, swordtail fish have testis-specific expression of a PRDM9β type gene that lacks the KRAB and SSXRD domains, and has a slowly evolving ZF array (*Figure 4*; *Figure 4—figure supplement 1*); they further carry substitutions at two catalytic residues of the SET domain (Y341F and Y357P), as well as at residues of the SET domain implicated in H3K4me2 recognition (see Materials and methods). Based on these features, we predicted that they should behave like a PRDM9 knockout, with no increase in recombination around the PRDM9 motif.

To test these predictions, we collected ~1X genome coverage from 268 natural hybrids and inferred crossover events from ancestry switchpoints between the two parental species using a hidden Markov model (see Materials and methods). By this approach, we found recombination rates to

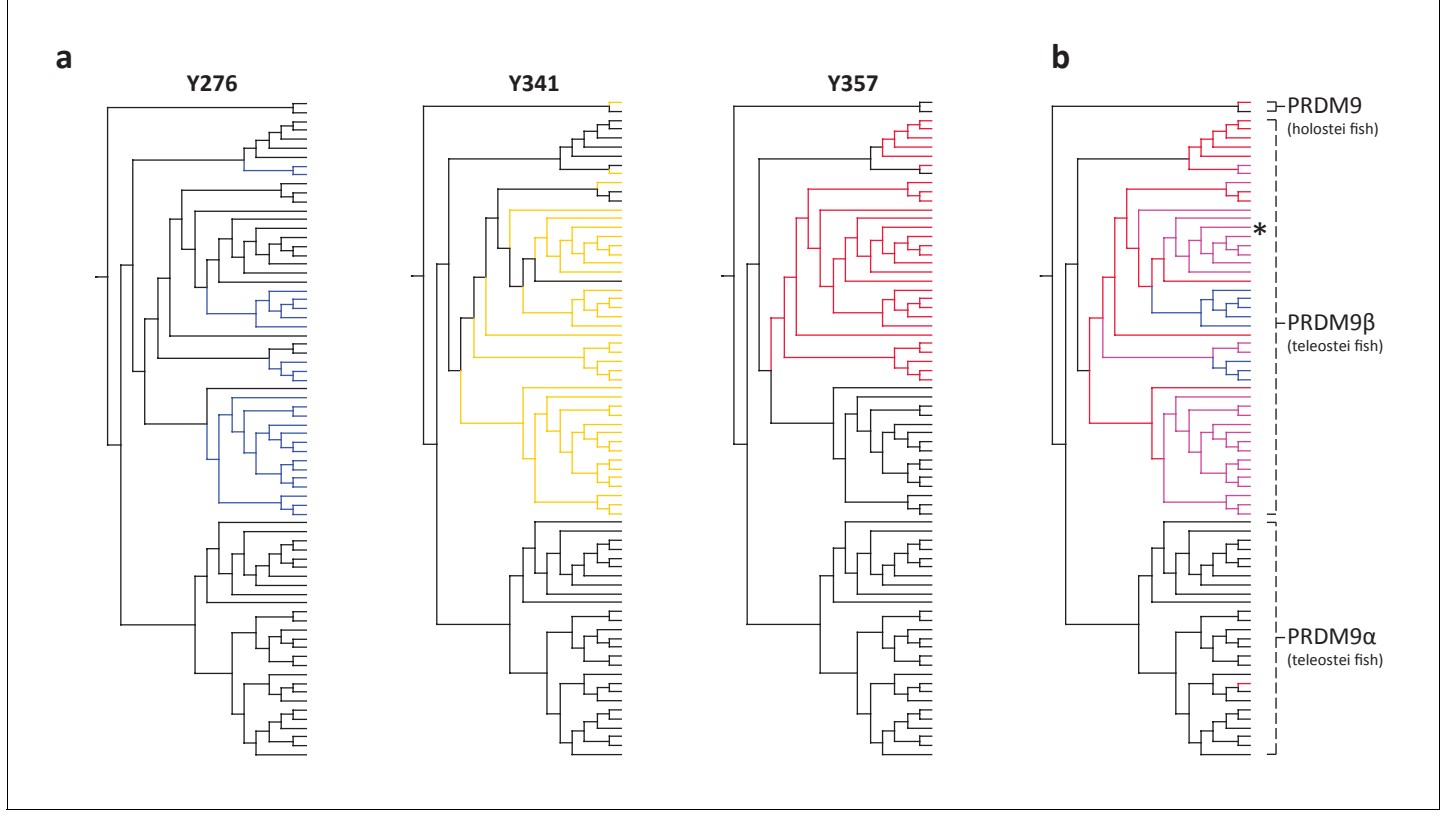

**Figure 3.** Substitutions at SET domain catalytic residues in bony fish PRDM9 genes. (a) Lineages within bony fish carrying substitutions at each of three tyrosine residues involved in H3K4me3 catalysis in human PRDM9 are shown in blue, yellow and red. (b) Lineages carrying substitutions at one, two or three of these residues are shown in red, pink and blue respectively. All PRDM9β genes as well as a partial PRDM9 ortholog from holostean fish carry one or more substitutions at these residues. The PRDM9β gene from *Xiphophorus* is indicated by the presence of asterisk.

be elevated near TSSs and CGIs, two promoter-like features (*Figure 4*; *Figure 4—figure supplement 2*). Moreover, and in contrast to what is observed in species with PRDM9-mediated recombination (*Figure 4—figure supplement 3*), there is no elevation in recombination rates near computationally-predicted PRDM9 binding sites (*Figure 4F*). These patterns resemble those previously reported for birds lacking PRDM9 (*Singhal et al., 2015*).

In addition, we performed native chromatin Chip-seq with an H3K4me3 antibody in *X. birchmanni* testis and liver tissue. Consistent with a role for H3K4me3 in inducing DSBs, recombination is increased around H3K4me3 peaks (testing the association with distance, rho = −0.072, p=2.3e-69; *Figure 4*), an effect that remains significant after correcting for distance to TSSs and CGIs (rho = −0.026, p=5.4e-10). In fact, the increase in recombination rate near the TSS is almost completely explained by the joint effects of proximity to H3K4me3 peaks and CGI (TSS with both: rho = −0.009, p=0.02). Windows that contain testis-specific H3K4me3 peaks have significantly higher recombination rates than those that contain liver-specific H3K4me3 peaks (*Figure 4; Figure 4—figure supplement 4*). However, H3K4me3 peaks in the testis are not enriched for the computationally predicted PRDM9 motifs (*Figure 4*), nor do they overlap with PRDM9 motifs in the testis more than the liver (see Methods). Conversely, sequence motifs associated with testis-specific H3K4me3 peaks do not resemble the predicted PRDM9 motif (*Figure 4; Figure 4—figure supplement 5*). Thus, there is no evidence that PRDM9 lays down the H3K4me3 marks associated with an increase in recombination in swordtails.

## Recombination landscapes in vertebrates with and without PRDM9
To put the genomic patterns of recombination in swordtail fish in an explicit comparative framework, we re-analyzed patterns of recombination near TSSs and CGIs in previously published genetic maps

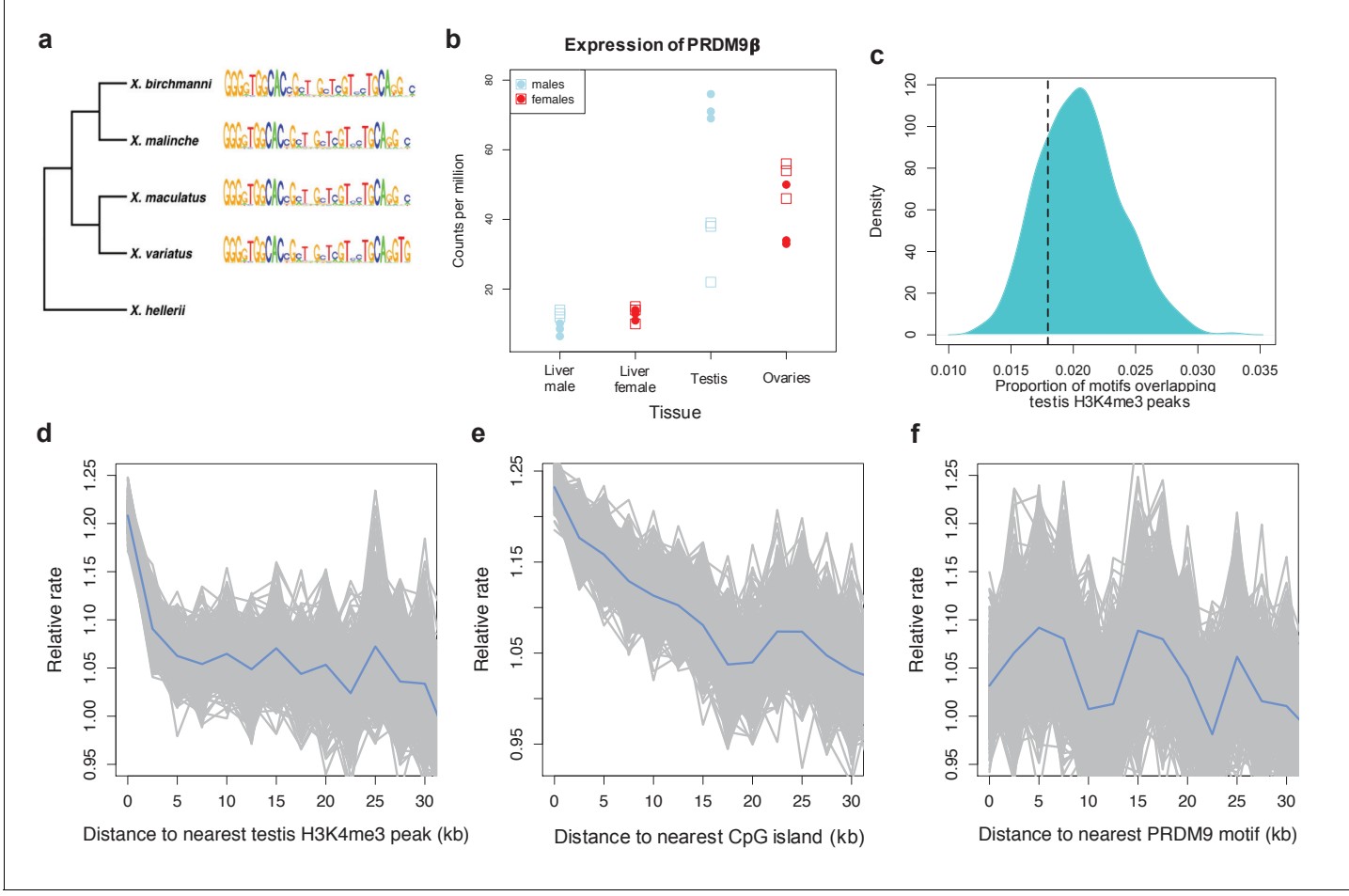

**Figure 4.** Patterns of recombination and PRDM9 evolution in swordtail fish. (**a**) The ZF array of PRDM9 appears to be evolving slowly in *Xiphophorus*, with few changes over 1 million years of divergence (***Cui et al., 2013***; ***Jones et al., 2013***). (**b**) PRDM9 is upregulated in the germline relative to the liver in *X. birchmanni* (circles) and X. *malinche* (squares; panel shows three biological replicates for each species). (**c**) The computationally-predicted PRDM9 binding sites is not unusually associated with H3K4me3 peaks in testes (**d**) Crossover rates increase near H3K4me3 peaks in testis (**e**) Crossover rates increase near CGIs (**f**) Crossover rates do not increase near computationally-predicted PRDM9 binding sites (see ***Figure 4—figure supplement 3*** for comparison). Crossover rates were estimated from ancestry switchpoints in naturally occurring hybrids between *X. birchmanni* and *X. malinche* (see Materials and methods).

The following figure supplements are available for figure 4:

**Figure supplement 1.** Expression levels of meiosis-related genes in swordtail fish tissues.

**Figure supplement 2.** Recombination frequency in swordtails as a function of distance to the TSS.

**Figure supplement 3.** Recombination rates show a peak near the computationally predicted PRDM9A binding motif in humans and gor-1 allele in gorillas.

**Figure supplement 4.** Higher observed recombination rate at testis-specific H3K4me3 peaks than liver-specific H3K4me3 peaks.

**Figure supplement 5.** MEME prediction of sequences enriched in testis-H3K4me3 peaks relative to liver-specific H3K4me3 peaks.

based on LD data from three species without functional PRDM9 genes (dog, zebra finch and long-tailed finch) and three species known to use PRDM9-mediated recombination (human, gorilla and mouse), as well as using a pedigree-based genetic map for one species with a complete PRDM9

ortholog, but for which no direct evidence of PRDM9's role in recombination has yet been reported (sheep; see Materials and methods for details and references).

Among species with complete PRDM9 genes, recombination rates are either weakly reduced near TSSs and CGIs or similar to what is seen in nearby windows (*Figure 5*; see *Figure 5—figure supplement 1* for results with genetic maps based on pedigrees or admixture switches instead of LD data in humans and dogs). In contrast, in all species lacking PRDM9 and in swordtail fish, the recombination rate is notably increased in windows overlapping either a TSS or CGI relative to nearby windows. Quantitative comparisons are difficult because of the varying resolution of the different genetic maps. Nonetheless, these results indicate that patterns of recombination near TSSs

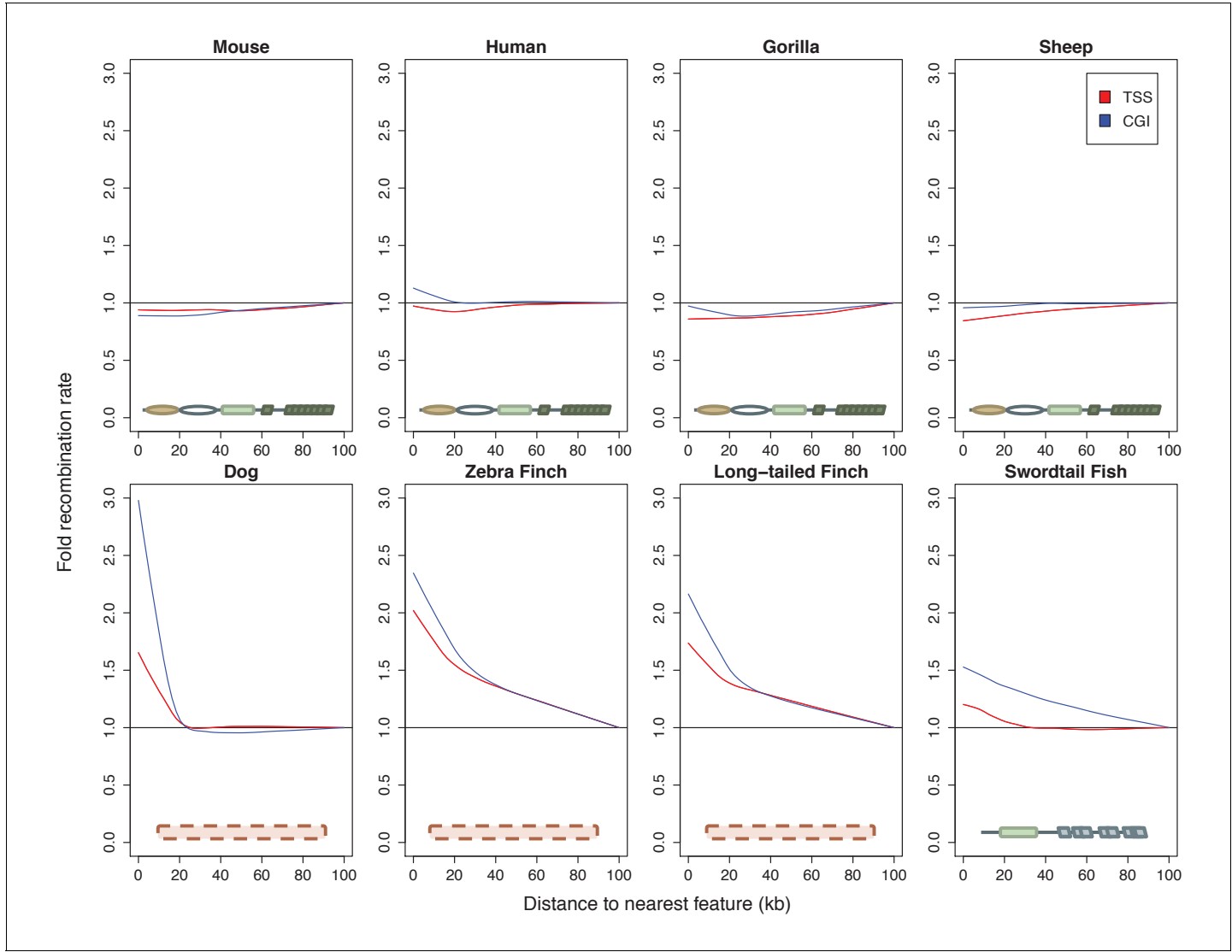

**Figure 5.** Patterns of recombination near TSSs and CGIs in species with and without complete PRDM9 orthologs. For each species, recombination rates were binned in 10 kb windows along the genome; curves were fit using gaussian loess smoothing. The fold change in recombination rates shown on the y-axis is relative to recombination rates at the last point shown. Species shown in the top row have complete PRDM9 orthologs (mouse, human, gorilla and sheep), whereas species in the bottom row have no PRDM9 ortholog (dog, zebra finch, long-tailed finch), or a partial PRDM9 ortholog (swordtail fish).

The following figure supplement is available for figure 5:

**Figure supplement 1.** Dependence of patterns of recombination near TSSs and CGIs in dog and human on the type of genetic map.

and CGIs differ between species carrying complete PRDM9 orthologs and species lacking PRDM9 altogether, and that swordtail fish exhibit patterns of recombination similar to species that completely lack PRDM9, despite the presence of a partial ortholog.

## Discussion

Based on our reconstruction of 227 PRDM9 orthologs across the vertebrate phylogeny, we inferred that the ancestral domain architecture of PRDM9 consisted of KRAB, SSXRD and SET domains followed by a C2H2 ZF array, and that this complete architecture was likely already in place in the common ancestor of vertebrates.

Moreover, even though to date only the functions of the SET domain and C2H2 ZF array have been connected to the role of PRDM9 in directing recombination, the evolutionary patterns uncovered here suggest that all four domains are important. The first line of evidence is that there is no observation of rapid evolution of the ZF domains in PRDM9 orthologs from which KRAB and SSXRD domains have apparently been lost (including a subset of species in which the catalytic activity of the SET domain is seemingly conserved), suggesting that there has not been rapid evolution of binding specificity. In contrast, there is evidence of rapid evolution of the PRDM9 ZF in all species that have KRAB, SSXRD, SET, and ZF domains. Since plausible models suggest that the rapid evolution of PRDM9 binding affinity is a consequence of the role of this gene in directing recombination (see Introduction), this observation suggests that all four domains are required for this role.

The second piece of evidence is that swordtail fish with a truncated copy of PRDM9 that is missing KRAB and SSXRD domains behave like PRDM9 knockouts in their fine-scale recombination patterns. It is unclear if this behavior can be attributed to loss of the N terminal domains, since two key catalytic residues within the SET domain were also substituted in this species. However, substitutions at catalytic residues are only seen in PRDM9 genes that have lost KRAB and/or SSXRD domains or have lost the ZF entirely. When the ZF is lost, PRDM9 obviously cannot induce DSBs by binding DNA and its new role may not require the same methyltransferase specificities. We speculate that the absence of KRAB and SSXRD domains in a PRDM9 ortholog may similarly signify that PRDM9 is no longer used to direct recombination and lead to reduced constraint on the catalytic activities of the SET domain. Consistent with this hypothesis, a recent paper suggests that the KRAB domain may play a role in recruiting the recombination machinery (*Parvanov et al., 2017*).

If the partial ortholog of PRDM9 is not used to direct recombination at all, then the overall conservation of the protein points to another role of the gene. In that regard, we note that partial PRDM9 orthologs share their domain architecture with other members of the PRDM gene family, many of which act as transcription factors (*Hayashi et al., 2005*; *Vervoort et al., 2016*).

Conversely, if the presence of all four domains, conservation of catalytic residues, and the rapid evolution of the ZF array are sufficient indications of PRDM9-directed recombination, then the role of PRDM9 in directing recombination appears to have originated before the diversification of vertebrates. It would follow that many non-mammalian vertebrate species, such as snakes, use the gene to determine the location of recombination hotspots. One hint in that direction is provided by the high allelic diversity seen in the ZF within a python species (*Python bivittatus*), reminiscent of patterns observed in apes (*Schwartz et al., 2014*; *Figure 1—figure supplement 5*). Assessing the role of PRDM9 in directing recombination in these species is a natural next step in understanding the evolution of recombination mechanisms.

It further appears that the intact PRDM9 has often been duplicated, with more than one copy associated with recombination rate variation in cattle (*Sandor et al., 2012*; *Ma et al., 2015*). Based on the RAxML tree of the SET domain, we count 55 independent cases of duplications. How commonly more than one copy of PRDM9 retains a role in directing recombination remains to be investigated.

More generally, the distribution of PRDM9 across vertebrates raises the question of why species switch repeatedly from one recombination mechanism to another. Although PRDM9-directed recombination clearly confers enough of an advantage for it to be widely maintained in vertebrates, at least six taxa of vertebrates carry only partial PRDM9 orthologs and the gene has been lost entirely at least three times (based on 227 orthologs; *Figure 1*, *Figure 2*). Thus, PRDM9 is not essential to meiotic recombination in the sense that SPO11 is, for example (*Lam and Keeney, 2014*). Instead, the role of PRDM9 is perhaps best envisaged as a classic, trans-acting recombination rate modifier

(*Otto and Barton, 1997*, *Otto and Lenormand, 2002*; *Coop and Przeworski, 2007*), which was favored enough to be adopted at some point in evolution, but not so strongly or stably as to prevent frequent losses.

In this regard, it is worth noting that in mammalian species studied to date, recombination rates are lower near promoters than in species lacking PRDM9 (*Myers et al., 2005*; *Coop et al., 2008*). Because recombination hotspots have higher rates of point mutations, insertions and deletions, and experience GC-biased gene conversion, there may be an advantage conferred by directing recombination to non-genic regions. Recombination at the TSS could have the further disadvantage of uncoupling coding and regulatory variants, potentially uncovering negative epistasis, and therefore leading to indirect selection for decreased recombination at the TSS. Alternatively (but non mutually-exclusively), because PRDM9 binding motifs are strongly associated with certain transposable element classes in mammals (*Myers et al., 2008*), the role of PRDM9 in recombination could be related to the regulation of certain families of transposable elements. With a more complete picture of recombination mechanisms and their consequences across the tree of life, these hypotheses can start to be tested in an evolutionary context.

## Materials and methods

### Identification of putative PRDM9 orthologs from the RefSeq database

As a first step in understanding the distribution of PRDM9 in vertebrates, we identified putative PRDM9 orthologs in the RefSeq database. We used the *blastp* algorithm (*Altschul et al., 1990*) using the *Homo sapiens* PRDM9 sequence, minus the rapidly evolving tandem ZF array, with an e-value threshold of 1e-5. We downloaded GenPept files and used Batch Entrez to retrieve the corresponding GenBank files (September 2016). The longest transcript for each locus and amino acid and DNA sequences corresponding to the KRAB, SSXRD and SET domains of these sequences (as annotated by the Conserved Domain Database; *Marchler-Bauer et al., 2015*), were downloaded using a R script (*Supplementary file 4*). The retrieved SET domain sequences, an additional 44 retrieved from whole genome assemblies, as well as seven retrieved from RNAseq datasets for five species without sequenced genomes (see Predicting PRDM9 orthologs from whole genome sequences) were input into ClustalW2 (*Larkin et al., 2007*), in order to generate a neighbor-joining (NJ) guide tree (see *Figure 1—figure supplement 2*). This approach was used to identify and remove genes that cluster with known PRDM family genes from humans and that share the SET domain of PRDM9 but were previously reported to have diverged from PRDM9 before the common ancestor of vertebrates (*Vervoort et al., 2016*; see Phylogenetic Analysis of PRDM9 orthologs and related gene families).

### Predicting PRDM9 orthologs from whole genome sequences

There were a number of groups not included in the RefSeq database or for which we were unable to identify PRDM9 orthologs containing the complete domain architecture. For 33 representative species from these groups, we investigated whether we could find additional PRDM9 orthologs in their whole genome assemblies (see *Supplementary files 1,2*) (*Song et al., 2015*; *Xiong et al., 2016*; *Bradnam et al., 2013*; *Georges et al., 2015*). To this end, we ran *tblastn* against the whole genome assembly, using the PRDM9 ortholog from the most closely related species that contained a KRAB domain, a SET domain, and at least one ZF domain (*Supplementary file 1B*). The number of hits to each region was limited to ten, and gene models were only predicted when a *blast* hit to the SET domain was observed with an e-value threshold of 1e-10.

When a single contig was identified containing an alignment to the full length of the query sequence, this contig was input into Genewise, along with the PRDM9 protein sequence from a species with a high quality ortholog (using a closely related species where possible), in order to create a new gene model. When PRDM9 domains were found spread across multiple contigs, we needed to arrange them in order to generate the proper sequences of the genomic regions containing PRDM9 orthologs from each species. When linkage information was available and we observed the presence of PRDM9 domains on linked contigs, we arranged the sequences of these contigs accordingly, with gaps padded with 100 Ns, before inputting them into Genewise. In cases where linkage information was not available, our approach differed depending on whether or not we identified more than one

hit to each region of the query sequence. In species where there appeared to be only one PRDM9 ortholog, we arranged the contigs according to the expected arrangements of the domains, though did not include any ZF arrays unless they were found on the same contig as the complete SET domain because the repeat structure of these domains makes homology difficult to infer. In species with more than one PRDM9 ortholog, we did not attempt to construct any gene models not supported by linkage or by transcripts identified from the same species (see Confirming the expression or absence of PRDM9 in the testes of major phylogenetic groups; *Supplementary file 1B* for details).

The positions of KRAB, SSXRD and SET domains for each gene model were annotated using *CD-blast* (Domain Accessions smart00317, pfam00856, cl02566, pfam09514, pfam01352, cd07765, smart00349). This approach resulted in the identification of additional PRDM9 orthologs containing at minimum the SET domain, in two jawless fish, two cartilaginous fish, nine bony fish, one monotreme, two marsupials, one turtle, four lizards, and eight snakes (*Supplementary file 1A*). We were unable to detect PRDM9 orthologs in one lizard (*Anolis carolinenesis*), or in any of three amphibian species (*Supplementary file 1B*). We used RNA-seq data to investigate whether these negative findings are due to genome assembly quality or reflect true losses (see below).

## Phylogenetic analysis of PRDM9 orthologs and related gene families

To understand the evolution of PRDM9 within vertebrates, we used a phylogenetic approach. We first built an alignment of the amino acid sequences of putative PRDM9 and PRDM11 SET domains using Clustal Omega (*Sievers et al., 2011*). We included genes clustering with PRDM11 because it had been reported that PRDM11 arose from a duplication event of PRDM9 in the common ancestor of bony fish and tetrapods (*Vervoort et al., 2016*), and we were interested in identifying any PRDM9 orthologs carried by vertebrate species that may precede this duplication event. The alignment coordinates were then used to generate a nucleotide alignment, which was used as input into the program RAxML (v7.2.8; *Stamatakis, 2006*). We performed 100 rapid bootstraps followed by maximum likelihood estimation of the tree under the General Reversible Time substitution model, with one partition for each position within a codon. The resulting phylogeny contained monophyletic groups corresponding to the PRDM9 and PRDM11 duplication event, with 100% bootstrap support (*Figure 1—figure supplement 1*). These groups were used to label each putative ortholog as PRDM9 or PRDM11. Only jawless fish have PRDM9 orthologs basal to this duplication event, suggesting PRDM11 arose from PRDM9 before the common ancestor of cartilaginous fish and bony vertebrates. We observed at least one PRDM11 ortholog in each of the other vertebrate species examined.

Within teleost fish, we identified two groups of PRDM9 orthologs, which we refer to as PRDM9$\alpha$ and PRDM9$\beta$ (*Figure 2—figure supplement 1*). While the bootstrap support for the monophyly of the two groups is only 75% for PRDM9$\alpha$ and 54% for PRDM9$\beta$, the potential duplication event suggested by this tree is coincident with the whole genome duplication event known to have occurred in the common ancestor of teleost fish (*Taylor et al., 2003*). Moreover, the phylogenetic grouping based on the SET domain is concordant with general differences in the domain architectures between the two orthologs: In contrast to PRDM9$\alpha$, PRDM9$\beta$ genes have derived ZF array structures, containing multiple tandem ZF arrays spread out within the same exon (*Figure 2—figure supplement 1*) and are always found without the KRAB and SSXRD domains, whereas PRDM9$\alpha$ genes generally have a single tandem array of ZFs consistent with the inferred ancestral domain architecture, and occasionally have KRAB and SSXRD domains (*Figure 2*).

## Confirming the expression or absence of PRDM9 in the testes of major phylogenetic groups

A necessary condition for PRDM9 to be involved in recombination is its expression in meiotic cells. For groups of taxa in which we detected a PRDM9 ortholog, we evaluated whether this ortholog was expressed in the testes, using a combination of publically available RNAseq data and RNAseq data that we generated. Additionally, in groups of species where PRDM9 appeared to be absent from the genome, we used publically available RNAseq data to confirm the absence of expression of PRDM9. In both cases, we used a stringent set of criteria to try to ensure that the absence of expression was not due to data quality issues (see details below).

We downloaded data for jawless fish, cartilaginous fish, bony fish, coelacanth, reptile, marsupial and monotreme species for which Illumina RNAseq data were available (*Supplementary file 2A,C, D*). We additionally generated RNAseq data for two percomorph fish species, *Xiphophorus birchmanni* and *X. malinche* (see below). Downloaded reads were converted to fastq format using the sratoolkit (v2.5.7; *Leinonen et al., 2011*) and trimmed for adapters and low quality bases (Phred <20) using the program cutadapt (v1.9.1; https://cutadapt.readthedocs.io/en/stable/). Reads shorter than 31 bp post-quality trimming were discarded. The program interleave_fastq.py was used to combine mate pairs in cases where sequence data were paired-end (*Crawford, 2014*; https://gist.github.com/ngcrawford/2232505). De-novo transcriptome assemblies were constructed using the program velvet (v1.2.1; *Zerbino and Birney, 2008*) with a kmer of 31; oases (*Schulz et al., 2012*; v0.2.8) was used to construct transcript isoforms. Summaries of these assemblies are available in *Supplementary file 2A*.

In order to identify potential PRDM9 transcripts in each of 24 assembled transcriptomes, we implemented *tblastn* using the human PRMD9 sequence, minus the ZF domain, as the query sequence, with an e-value threshold of 1e-5. The identified transcripts were extracted with a custom script and blasted to our dataset of all PRDM genes (*Supplementary file 5*). If the best blast hit was a PRDM9 ortholog, we considered PRDM9 expression in the testis to be confirmed (see results in *Supplementary file 2C*). For five species lacking genome assemblies, we extracted PRDM9 orthologs with best *blast* hits to human PRDM9/7 and included these in our phylogenetic analyses (see Phylogenetic Analysis of PRDM9 orthologs and related gene families).

Failure to detect PRDM9 could mean that PRDM9 is not expressed in that tissue, or that data quality and sequencing depth are too low to detect its expression. To distinguish between these possibilities, we used other recombination-related genes as positive controls, reasoning that if expression of several other conserved recombination-related genes were detected, the absence of PRDM9 would be more strongly suggestive of true lack of expression. Eight recombination-related genes are known to be conserved between yeast and mice (*Lam and Keeney, 2014*). We used the subset of seven that could be reliably detected in whole genome sequences, and we asked which transcriptomes had reciprocal best *tblastn* (e-value <1e-5) hits to all of these proteins, using query sequences from humans (*Supplementary file 2A*; *Supplementary file 2D*). In addition, in order to assess whether PRDM9 expression might simply be lower than that of other meiotic genes, we quantified absolute expression of PRDM9 and the seven conserved recombination-related proteins in whole testes, using data from three major taxa (bony fish, mammals, and reptiles); see Analysis of PRDM9 expression levels and expression levels of other conserved recombination-related genes for more details. Together, these results suggest that not detecting PRDM9 in whole testes transcriptomes provides support for its absence.

## RNA extraction and sequencing of liver and gonad tissue from swordtail fish

Three *Xiphophorus birchmanni* and three *X. malinche* were collected from the eastern Sierra Madre Oriental in the state of Hidalgo, Mexico. Fish were caught using baited minnow traps and were immediately euthanized by decapitation (Texas A&M AUP# - IACUC 2013–0168). Testis, ovaries, and liver were dissected and stored at 4°C in RNAlater. Total RNA was extracted from testis, ovary and liver tissue using the Qiagen RNeasy kit (Valencia, CA, USA) following the manufacturer's protocol. RNA was quantified and assessed for quality on a Nanodrop 1000 (Nanodrop technologies, Willmington, DE, USA) and approximately 1 µg of total RNA was used input to the Illumina TruSeq mRNA sample prep kit. Samples were prepared following the manufacturer's protocol with minor modifications. Briefly, mRNA was purified using manufacturer's beads and chemically fragmented. First and second strand cDNA was synthesized and end repaired. Following A-tailing, each sample was individually barcoded with an Illumina index and amplified for 12 cycles. The six libraries were sequenced on the HiSeq 2500 at the Lewis Sigler Institute at Princeton University to collect single-end 150 bp reads, while single-end 100 bp data was collected on the HiSeq 4000 at Weill Cornell Medical College for all other samples (SRA Accessions: SRX2436594 and SRX2436597). Reads were processed and a de novo transcriptome assembled for the highest coverage testis library following the approach described above for publicly available samples. Details on assembly quality are available in *Supplementary file 2A*. Other individuals were used in analysis of gene expression levels (see next section).

## Analysis of PRDM9 expression levels and expression levels of other conserved recombination-related genes

To determine whether some of the genes in our conserved recombination-related gene set were expressed at similar levels to PRDM9, implying similar detection power, we examined expression levels of these genes in three species representing the bony fish, reptilian, and mammalian taxa (*Xiphophorus malinche*, *Pogona vitticeps*, and *Homo sapiens*).

To quantify expression in *X. malinche*, we mapped trimmed reads from testes RNAseq libraries that we generated from three individuals to the *X. maculatus* reference genome (v4.4.2; *Schartl et al., 2013*; *Amores et al., 2014*) using bwa (v0.7.10; *Li and Durbin, 2009*). The number of trimmed reads per individual ranged from 9.9 to 27.5 million. We used the program eXpress (v1.5.1; *Roberts et al., 2011*) to quantify fragments per kilobase of transcript per million mapped reads (FPKM) for each gene, and extracted the genes of interest from the results file based on their ensembl gene id. eXpress also gives confidence intervals on its estimates of FPKM.

For the bearded lizard *Pogona vitticeps*, we only had access to one publically available testis-derived RNAseq library. We followed the same steps used in analysis of swordtail FPKM except that we mapped to the transcriptome generated from the data and identified transcripts belonging to recombination-related gene sets using the reciprocal best *blast* hit approach described above.

Several publically available databases already exist for tissue specific expression in humans. We downloaded the 'RNA gene dataset' from the Human Protein Atlas (v15, http://www.proteinatlas.org/about/download). This dataset reports average FPKM by tissue from 122 individuals. We extracted genes of interest from this data file based on their Ensembl gene id.

Examination of these results demonstrated that other meiotic genes (2-5) in each species had expression levels comparable to PRDM9 (*Figure 1—figure supplement 3*). This finding suggests that these genes are appropriate positive controls, in that detecting their expression but not that of PRDM9 provides evidence against expression of PRDM9 in testes.

## Confirmation of PRDM9 domain loss and investigation of loss of function

In addition to complete losses of PRDM9, we were unable to identify one or more functional domains of PRDM9 in orthologs identified from the platypus, Tasmanian devil, elephant shark, all bony fish and several placental mammals.

To ask whether the missing PRDM9 domains were truly absent from the genome assembly, we first used a targeted genome-wide search. To this end, we performed a *tblastn* search of the genome against the human PRDM9 ortholog with an e-value of 1e-10. For all *blast* hits, we extracted the region and 2 Mb flanking in either direction, translated them in all six frames (http://cgpdb.ucdavis.edu/DNA_SixFrames_Translation/), and performed an *rpsblast* search of these regions against the CDD (database downloaded from NCBI September 2016) with an e-value of 100 to identify any conserved domains, even with weakly supported homology. We extracted all *rpsblast* hits to the missing functional domain (SET CDD id: smart00317, pfam00856, cl02566; SSXRD CDD id: pfam09514; KRAB domains pfam01352, cd07765, smart00349) and used them as query sequences in a *blastp* search against all KRAB, SSXRD and SET containing proteins in the human genome. If PRDM9 or PRMD7 was the top *blast* hit in this search, we considered that the missing domain could be a result of assembly or gene model prediction error (if not, we investigated the potential loss of these domains further). This approach allowed us to rule out genome-wide losses of PRDM9 domains in nine out of 14 species of mammals for which our initial approach had failed to identify complete PRDM9 orthologs. In each case, we checked whether or not the identified domains were found adjacent to any of our predicted gene models and adjusted the domain architecture listed for these RefSeq genes accordingly in our dataset (see *Supplementary file 1A*). In five species of mammals (Tasmanian devil, three bat species, and the aardvark), we only identified a partial PRDM9 ortholog, but we were unable to confirm the loss of domains using RNAseq data (see next section). Within bats, each partial gene model starts within 500 bp of an upstream gap in the assembly. Moreover, we were able to identify a KRAB domain corresponding to PRDM9 from a closely related species of bat (*Myotis brandtii*). Thus, we believe that in the case of bats, these apparent domain losses are due to assembly errors or gaps.

For species with available RNAseq data from taxa in which we predicted PRDM9 N-terminal truncation based on our initial analyses, we sought to confirm the domain structure observed in the genome with de novo transcriptome assemblies from testis RNAseq (described above). As before, we only considered transcriptomes that passed our basic quality control test (*Supplementary file 2D*). Because RNAseq data are not available for all species with genome assemblies, we were only able to perform this stringent confirmation in a subset of species (*Supplementary file 2C*). As a result, we consider cases where N-terminal losses are confirmed in the genome as possible losses but are most confident about cases where N-terminal losses are observed both in the genome and transcriptome.

To examine the transcripts of PRDM9 orthologs from the transcriptome assemblies (*Supplementary file 2A*), for each domain structure, we translated each transcript with a *blast* hit to the human PRDM9 in all six frames and used *rpsblast* against all of these translated transcripts, with an e-value cutoff of 100 (as described above). Finally, we performed a reciprocal nucleotide *blast* (*blastn*; e-value cutoff 1e-20) to confirm that these transcripts were homologous to the PRDM9 ortholog identified using phylogenetic methods in these taxa. Results of this analysis can be found in *Supplementary file 2C*. In summary, there were two cases where the transcriptomes supported additional domain structures not found in the whole genome sequence (*Supplementary file 2C*): a PRDM9 ortholog from the spotted gar (*Lepisosteus oculatus*) that was observed to have a KRAB domain not identified in the genome sequence, and a PRDM9α ortholog from the Atlantic salmon (*Salmo salar*) that was observed to have both KRAB and SSXRD domains not identified in the genome search. In all other cases, we confirmed the losses of either the KRAB or SSXRD domains, including: (i) PRDM9β orthologs missing KRAB and SSXRD domains in all species of teleost fish expressing these orthologs (*Supplementary file 2B,C*) (ii) PRDM9α orthologs missing KRAB and SSXRD domains identified from *Astyanax mexicanus, Esox lucius, Gadus morhua,* and *Ictalurus punctatus*, and (iii) loss of the KRAB domain from one PRDM9 ortholog in monotremata (*O. anatinus*) and both KRAB and SSXRD domains from the other ortholog in this species.

For all groups in which we confirm that there is only a partial PRDM9 ortholog based on the above analyses, we asked whether the PRDM9 gene in question has likely become a pseudogene (as it has, for example, in canids; *Oliver et al., 2009*; *Muñoz-Fuentes et al., 2011*), in which case the species can be considered a PRDM9 knockout. Though such events would be consistent with our observation of many losses of PRDM9, they would not be informative about the role of particular PRDM9 domains in recombination. For this analysis, we aligned the SET domain of the PRDM9 coding nucleotide sequence to a high-quality PRDM9 sequence with complete domain structure from the same taxon using Clustal Omega (see *Supplementary file 3A*), except for the case of PRDM9β in bony fish and the PRDM9 ortholog from cartilaginous fish, where such a sequence was not available. In the case of PRDM9β, we compared the sequence between *X. maculatus* and *A. mexicanus*, sequences that are >200 million years diverged (*Hedges et al., 2015*). In the case of cartilaginous fish, we used the sequence from *R. typus* and *C. milii*, which are an estimated 400 million years diverged (*Hedges et al., 2015*).

We analyzed these alignments with codeml, comparing the likelihood of two models, one with a fixed omega of 1 and an alternate model without a fixed omega, and performed a likelihood ratio test. A significant result for the likelihood ratio test provides evidence that a gene is not neutrally-evolving (*Supplementary file 3A*). In all cases of N-terminal truncation analyzed, dN/dS is significantly less than one (*Supplementary file 3A*). While it is possible that some of these cases represent very recently pseudogenized genes, the widespread evidence for purifying selection on the SET domain strongly suggests that these PRDM9 orthologs are functionally important.

We also investigated constraint in all mammalian Ref-seq orthologs that appear to lack only an annotated KRAB or SSXRD domain; for this larger number of genes, we did not confirm all domain losses, due to the large number of genome searches that would be required and lack of RNAseq data for most species. We found evidence of purifying selection in all cases except for five PRDM7 orthologs from primates, for which we had been unable to identify a KRAB domain (*Supplementary file 3C*). PRDM7 is thought to have arisen from a primate specific duplication event and to have undergone subsequent losses of the C2H2 ZF array and of some catalytic specificity of its SET domain (*Blazer et al., 2016*). Thus, PRDM7 orthologs are unlikely to function in directing recombination. Our findings further suggest they are evolving under very little constraint, and may even be non-functional. More generally, within placental mammals, the majority of partial PRDM9

orthologs that we identified lack the ZF array completely or have truncated arrays (notably, there are fewer than four tandem ZFs in 24 of 28 orthologs), in sharp contrast to other taxa in which partial orthologs to PRDM9 lack the N terminal domains, yet have conserved ZF arrays and are constrained. Moreover, the paralogs lacking a long ZF tend to be found in species that already carry a complete PRDM9 ortholog (21 of 24). Thus, some of these cases may represent recent duplication events in which one copy of PRDM9 is under highly relaxed selection, similar to PRDM7 in primates.

## Evolutionary patterns in the SSXRD domain

The SSXRD domain is the shortest functional domain in the PRDM9 protein. One species of cartilaginous fish (*Rhincodon typus*), and several species of bony fish (*Anguilla anguilla, A. rostrata, A. japonica, Salmo trutta, S. salar*) have weakly predicted SSXRD domains (e-values >10, see *Supplementary file 1B* and *2C*). This observation is potentially suggestive of functional divergence or loss of this domain. Unfortunately, because the domain is so short, there is little power to reject dN/dS = 1; though the estimate of dN/dS was 0.10 and 0.11 between cartilaginous fish and eel and salmon orthologous regions, respectively, the difference between models was not significant in either case. Based on these findings, we tentatively treat the weakly predicted SSXRD domain in *Rhincodon typus* and in the above species of bony fish as evidence that this domain is present in these species, but note that we were unable to identify a similar region in predicted gene models from another species of cartilaginous fish (*Callorhinchus milii*).

## PCR and Sanger sequencing of python PRDM9

We performed Sanger sequencing of *Python bivittatus* PRDM9 from a single individual to collect additional data on within species diversity of the ZF array (*Figure 1—figure supplement 5*). Primers were designed based on the *Python bivattatus* genome (*Castoe et al., 2013*) to amplify the ZF containing exon of PRDM9 and through a gap in the assembly. Primers were assessed for specificity and quality using NCBI Primer Blast (http://www.ncbi.nlm.nih.gov/tools/primer-blast/) against the nr reference database and were synthesized by IDT (Coralville, IA, USA).

DNA was extracted from approximately 20 mg of tissue using the Zymo Quick-DNA kit (Irvine, CA, USA) following the manufacturer's protocol. PCR was performed using the NEB Phusion High-Fidelity PCR kit (Ipswich, MA, USA). Reactions were performed following manufacturer's instructions with 60 ng of DNA and 10 µM each of the forward (ZF: 5'TTTGCCATCAGTGTCCCAGT'3; gap: 5' GCTTCCAGCATTTTGCCAGTT'3) and reverse (ZF: 5' TTGATTCACTTGTGAGTGGACAT'3; gap: 5' GAGCTTTGCTGAAATCGGGT'3) primers. Products were inspected for non-specific amplification on a 1% agarose gel with ethidium bromide, purified using a Qiagen PCR purification kit (Valencia, CA, USA) and sequenced by GeneWiz (South Plainfield, NJ, USA).

## Analysis of PRDM9 ZF array evolution

In species in which PRDM9 is known to play a role in recombination, the level of sequence similarity between the individual ZFs of the tandem array is remarkably high, reflective of high rates of ZF turnover due to paralogous gene conversion and duplication events (*Oliver et al., 2009*; *Myers et al., 2010*; *Jeffreys et al., 2013*). It has further been observed that DNA-binding residues show high levels of amino acid diversity, suggestive of positive selection acting specifically at DNA-binding sites, that is, on binding affinity (e.g. *Oliver et al., 2009*; *Schwartz et al., 2014*). These signals have been previously studied by comparing site specific rates of synonymous versus non-synonymous substitutions (dN/dS) between paralogous ZFs in PRDM9's tandem ZF array (*Oliver et al., 2009*). Assessing statistical significance using this approach is problematic, however, because the occurrence of paralogous gene conversion across copies means that there is no single tree relating the different ZFs, in violation of model assumptions (*Schierup and Hein, 2000*; *Wilson and McVean, 2006*). Here, we used a statistic sensitive to both rapid evolution at DNA-binding sites and high rates of gene conversion: the total proportion of amino acid diversity observed at DNA-binding sites within the ZF array. We then assessed significance empirically by comparing the value of this statistic to other C2H2 ZF genes from the same species (where possible).

To this end, for each species with a PRDM9 ortholog, we downloaded the nucleotide and protein sequences for all available RefSeq genes with a C2H2 ZF motif annotated in Conserved Domain Database (pfam id# PF00096). To simplify alignment generation, we only used tandem ZF arrays

with four or more ZFs matching the 28 amino acid long C2H2 motif (X2-CXXC-X12-HXXXH-X5 where X is any amino acid). In all of our analyses, if a gene had multiple tandem ZF arrays that were spatially separated, only the first array of four or more adjacent ZFs was used for the following analysis (*Supplementary file 3B*). However, an alternative analysis using all ZFs or different subsets of ZFs led to qualitatively similar results for the PRDM9β orthologs from bony fish, where ZFs are commonly found in multiple tandem arrays separated by short linker regions in the predicted amino acid sequence (*Figure 2—figure supplement 1*; *Figure 2—figure supplement 2*). For species with PRDM9 orthologs with fewer than five ZFs, we implemented *blastn* against the whole genome sequence using the available gene model as a query sequence, in order to determine whether or not there was a predicted gap within the ZF array, and, if there was, to identify any additional ZFs found in the expected orientation at the beginning of the adjacent contig. This approach was able to successfully identify additional ZF sequences on contigs adjacent to PRDM9 in the genome assembly for two species (*Latimeria chalumnae* and *Protobothrops mucrosquamatus*). These ZFs were included in subsequent analysis (*Supplementary file 1A*).

Using the alignments generated above, we determined the amino acid diversity along the ZF domains of PRDM9 genes and all other C2H2 ZFs from the same species (*Table 1*, *Supplementary file 3B*), and calculated the proportion of the total amino acid diversity at canonical DNA-binding residues of the ZF array. Specifically, we calculated the heterozygosity $x_k$ at position k across the aligned ZFs from a single tandem array as:

$$x_k = 1 - \sum_{i=1}^{m} f_i^2$$

where m is the number of unique amino acids found at position k across the fingers, and $f_i$ is the frequency of the $i^{th}$ unique amino acid across the fingers. The total proportion $P$ of amino acid diversity assigned to DNA-binding residues is the sum of $x_k$ at DNA-binding sites over the sum of $x_k$ at all sites in the ZF array. To compare results to those for other genes, we ranked PRDM9 by the value $P$ compared to all other C2H2 ZF genes from the same species (*Table 1*, *Supplementary file 3B*).

We used the R package phylotools (*Zhang et al., 2012*; https://cran.r-project.org/web/packages/phylotools/index.html) to calculate a p-value for the correlation between complete domain structure and rapid evolution of the PRDM9 ZF array, taking into account phylogenetic relationships between PRDM9 orthologs. We coded these variables using a binary approach with '00' for incomplete domain structure and no evidence of rapid evolution and '11' for complete domain structure and evidence of rapid evolution. To describe the phylogenetic relationships between orthologs, we used the RAxML tree that we constructed from the SET domain for all PRDM9 orthologs. Species with missing ZF information, including species where PRDM9 has been lost, were excluded from this analysis using the drop.tip function of the ape package (*Paradis et al., 2004*), resulting in a tree with 91 tips. We used the phyloglm command to perform a logistic regression evaluating the relationship between domain structure and the odds of rapid evolution of the ZF array.

## Analysis of the SET domain catalytic residues

In order to investigate whether the catalytic function of the SET domain is conserved in the PRDM9 orthologs identified above, we asked whether any PRDM9 orthologs in our dataset carried substitutions at three catalytic residues shown to mediate the methyltransferase activity of human PRDM9 (*Wu et al., 2013*). To this end, we used Clustal Omega to create an amino acid alignment of the SET domain with 15 amino acids of flanking sequence for each PRDM9 ortholog in our dataset and asked whether the gene had substitutions to tyrosine residues at positions aligning to Y276, Y341 and Y357 in human PRDM9 (*Supplementary file 1A*). Domain alignments deposited at *Baker et al., 2017*.

In total, 57 genes were identified as having substitutions in at least one of these residues, including 11 from placental mammals and 46 from bony fish (*Supplementary file 1A*). To visualize the distribution of these substitution events within bony fish, we mapped these substitutions onto the phylogeny of PRDM9 orthologs generated above (*Figure 3*).

## Characterizing patterns of recombination in hybrid swordtail fish

Percomorph fish have a partial ortholog of PRDM9 that lacks the KRAB and SSXRD domains found in mammalian PRDM9. As a result, we hypothesized that they would behave like PRDM9 knockouts, in that the predicted PRDM9 binding motif would not co-localize with recombination events, and functional genomic elements such as the TSS and CGIs would be enriched for recombination events.

To build a hybrid recombination map, we generated low coverage sequence data for 268 individuals from a natural hybrid population ('Totonicapa') formed between the percomorph species *X. birchmanni* and *X. malinche* (RRID:SCR_008340) and sampled between 2013–2015. The two parental species are closely related, with pairwise sequence divergence <0.5% (*Schumer et al., 2014*). Interestingly, in sharp contrast to what is seen in placental mammals, the ZF is slowly evolving between *X. birchmanni* and *X. malinche* (dN/dS = 0.09; *Figure 4A*).

DNA was extracted from fin clips for the 268 individuals and libraries were prepared following *Stern, 2016*. Briefly, three to ten nanograms of DNA was mixed with Tn5 transposase enzyme precharged with custom adapters and incubated at 55 C for 15 min. The reaction was stopped by adding 0.2% SDS and incubating at 55 C for an additional seven minutes. One of 96 custom indices were added to each sample in a plate with an individual PCR reaction including 1 ul of the tagmented DNA; between 13–16 PCR cycles were used. After amplification, 5 ul of each reaction was pool and purified using Agencourt AMPpure XP beads. Library size distribution and quality was visualized on the Bioanalyzer 1000 and size selected by the Princeton Lewis Sigler Core Facility to be between 350–750 basepairs. Libraries were sequenced on the Illumina HiSeq 4000 at Weill Cornell Medical Center across three lanes to collect paired-end 100 bp reads.

Ancestry assignment in hybrids was performed using the Multiplexed Shotgun Genotyping ('MSG') pipeline (*Andolfatto et al., 2011*). This approach has been previously validated for genome-wide ancestry determination in late generation *X. birchmanni* x *X. malinche* hybrids (*Schumer et al., 2014*; *Schumer et al., 2016b*). Briefly, raw data was parsed by barcode and trimmed to remove low-quality basepairs (Phred quality score <20). Reads with fewer than 30 bp after trimming were discarded. Because of prohibitively long computational times, reads from individuals with more than 16 million reads were subsampled to 16 million before running the MSG pipeline. The minimum number of reads for an individual to be included was set to 300,000, since ancestry inference with fewer reads is predicted to have lower accuracy based on simulations (Schumer et al. 2015). This procedure resulted in 239 individuals for our final analysis, with an average coverage of 8.3 million reads, or ~1X genome-wide coverage.

The parameters used in the MSG run were based on previous work on this hybrid population (Schumer et al. in review). The expected number of recombination events per chromosome (recRate) was set to 8, based on a prior expectation of approximately 30 generations of admixture and assuming initial admixture proportions of 75% of the genome derived from *X. birchmanni* and 25% derived from *X. malinche*. Similarly, priors for each ancestry state were set based on these mixture proportions (par1 = 0.5625, par1par2 = 0.375 and par2 = 0.0625). The recombination rate scaling factor was set to the default value of 1.

Ancestry transitions were identified as the interval over which the posterior probability changed from ≥0.95 in support of one ancestry state to ≥0.95 for a different ancestry state. Breakpoint intervals that occurred within 10 kb of a contig edge were excluded from the analysis due to concerns that false breakpoints may occur more frequently near the edges of contigs. The identified recombination intervals varied significantly in their lengths, that is, in the resolution of the crossover event. The median resolution was 13 kb, with 75% of breakpoints resolved within 35 kb or less.

To evaluate the relationship between recombination frequency and genomic elements such as the TSS, CGIs, and computationally predicted PRDM9 binding sites, we needed to convert the observed recombination events into an estimate of recombination frequency throughout the genome. To this end, we considered the proportion of events observed in a particular 10 kb window; we note that this rate is not equivalent to a rate per meiosis. We filtered the data to remove windows within 10 kb of a contig boundary. Because the majority of events span multiple 10 kb windows, we randomly placed events that spanned multiple windows into one of the windows that they spanned.

We used the closest-feature command from the program bedops v2.4.19 (*Neph et al., 2012*) to determine the minimum distance between each 10 kb window and the functional feature of interest.

For the TSS, we used the Ensembl annotation of the *Xiphophorus maculatus* genome with coordinates lifted over to v.4.4.2 of the linkage group assembly (*Amores et al., 2014*; *Schumer et al., 2016a*) http://genome.uoregon.edu/xma/index_v1.0.php). For CGIs, we used the annotations available from the UCSC genome browser beta site (http://genome-test.cse.ucsc.edu/cgi-bin/hgTables?hgsid=391260460_COev5GTglYu74K2t24uaU4UcaTvP&clade=vertebrate&org=Southern+platy-yfish&db=xipMac1&hgta_group=allTracks&hgta_track=cpgIslandExt&hgta_table=0&hgta_region-Type=genome&position=JH556661%3A3162916-4744374&hgta_outputType=primaryTable&hgta_outFileName=). To identify putative PRDM9 binding sites, we used the ZF prediction software available at zf.princeton.edu with the polynomial SVM settings to generate a position weight matrix for the *X. malinche* and *X. birchmanni* PRDM9 orthologs (*Persikov and Singh, 2014*). This approach yielded identical predicted binding motifs in the two species (*Figure 4A*). We used this position weight matrix to search the *X. malinche* genome (*Schumer et al., 2014*) for putative PRDM9 binding sites with the meme-suite program FIMO (v4.11.1; *Grant et al., 2011*). We selected all regions with a predicted PRDM9 binding score of $\geq 5$. Since the individuals surveyed are interspecific hybrids, and the two species may differ in the locations of predicted PRDM9 binding sites, we repeated the FIMO search against the *X. birchmanni* genome, obtaining qualitatively identical results.

After determining the minimum distance between each 10 kb window and the features of interest, we calculated the average recombination frequency in hybrids as a function of distance from the feature of interest in 10 kb windows (*Figure 4*; *Figure 4—figure supplement 2*). To estimate the uncertainty associated with rates at a given distance from a feature, we repeated this analysis 500 times for each feature, bootstrapping windows with replacement. Because we found a positive correlation between distance from the TSS and CGIs in 10 kb windows with recombination frequency, we checked that power (i.e., the proportion of ancestry informative sites) was not higher near these features.

Most work in humans and mice has focused on the empirical PRDM9 binding motif rather than the computationally predicted motif. Since we expect the computationally predicted motif to be a poorer predictor of PRDM9 binding, we checked how its use would affect the analyses, by repeating the analysis described above for the computational prediction obtained for the human PRDM9A allele, using recombination rates in 10 kb windows estimated from the CEU LD map (*Frazer et al., 2007*; downloaded from: http://www.well.ox.ac.uk/~anjali/AAmap/). We also repeated this analysis for the gor-1 PRDM9 allele in *Gorilla gorilla*, using recombination rates in 10 kb windows estimated from a recent LD map (*Schwartz et al., 2014*; *Great Ape Genome Project et al., 2016*; downloaded from *Stevison, 2016*: https://github.com/lstevison/great-ape-recombination).

## Comparisons of recombination landscapes with and without PRDM9

To investigate whether patterns of recombination rates near the TSS and CGI systematically distinguish between species that do and do not use PRDM9-directed recombination, we compared available data across species. We downloaded previously published recombination maps for three species without PRDM9 genes (dog, *Auton et al., 2013*, zebra finch and long-tailed finch, *Singhal et al., 2015*) and four species with complete PRDM9 orthologs (human, *Frazer et al., 2007*; *Hinch et al., 2011*; gorilla, *Great Ape Genome Project et al., 2016*; sheep, *Johnston et al., 2016*; and mouse, *Brunschwig et al., 2012*).

For each species, we binned recombination rate into 10 kb windows along the genome, excluding the sex chromosomes and windows overlapping with assembly gaps from all analyses. For each species, we downloaded annotations of assembly gaps, TSSs and CGIs from the UCSC genome browser website. For CGI positions in the gorilla genome, we used the LiftOver tool (http://genome.ucsc.edu/cgi-bin/hgLiftOver) to convert the available coordinates for the GorGor4 genome assembly to the GorGor3 assembly. For zebra finch and long-tailed finches, we used the coordinates of CGIs and TSSs as annotated for the TaeGut3.2 genome assembly, noting that these coordinates are consistent with the TaeGut3.1 assembly for all chromosomes for which genetic distances were inferred in (*Singhal et al., 2015*).

For each map, we calculated the distance to the nearest TSS and to the nearest CGI by from the midpoint of each 10 kb window. To visualize these patterns, we fit a Gaussian loess curve using the distance to nearest TSS or CGI and recombination rate for each species, using only windows within 100 kb of a representative element. For visual comparison, we scaled the resulting curves by setting the y-value (recombination rate) of the last point to one.

A caveat is that other than for swordtail and sheep, we relied on LD based genetics maps, which estimate population recombination rates $4N_e r$, where $N_e$ is the effective population size and $r$ the recombination rate per meiosis. Because estimates of $N_e$ decrease near genes as a consequence of diversity-reducing linked selection (e.g., *Wright and Andolfatto, 2008*; *Hernandez et al., 2011*), a decrease in estimated population recombination rates near genes may not reflect a reduction in the recombination rate $r$. To explore the potential importance of this caveat, we considered two species where both LD maps and pedigree or admixture maps were available: dogs and humans. In both cases, the qualitative results were the same as for the LD-based maps (*Figure 5—figure supplement 1*). Since diversity-reducing linked selection should give rise, if anything, to a trough in diversity levels, it cannot explain the observed peaks at these features in species lacking PRDM9 or swordtail fish; in fact, since these species also experience this form of selection (e.g., *Singhal et al., 2015*), the true peaks in recombination rates near promoter-like features are likely somewhat more pronounced.

We note further that although the peak in recombination rate at these features in swordtail fish appears to be less prominent than in dog or birds, quantitative comparisons of different species are difficult because these maps differ in their resolution.

## Native chip-seq of *X. birchmanni* testis and liver tissue

Whole testis and liver were dissected from two *X. birchmanni* adults and stored in HypoThermosol FRS (BioLife Solutions, Bothell, WA) buffer on ice until processing. Native chromatin ChIP was performed as described previously (*Markenscoff-Papadimitriou et al., 2014*). Briefly, tissue was homogenized and lysed; the lysate was spun through a sucrose cushion (to pellet nuclei). Nuclei were resuspended in 500 ul MNase digestion buffer and digested with 1 unit of microccocal nuclease (MNase, Sigma N5386, St. Louis, MO) for 2 min at 37 C, then inactivated with 20 ul 0.5M EDTA and chilled on ice. The first soluble chromatin fraction was recovered by spinning for 10 min at 10,000 rcf at 4C and collecting the supernatant. To isolate the second soluble chromatin fraction, the pellet was resuspended in 500 µl dialysis buffer, rotated overnight at 4 C, then centrifuged for 10 min at 10,000 rcf at 4 C to pellet insoluble material. The digestion quality of each fraction was evaluated on an agarose gel. The two soluble fractions were combined for chromatin immunoprecipitation with 1 µg of H3K4me3 antibody (Millipore 04–745, Billerica, MA); 1/10 vol was retained as an input control. Antibody was bound to the remaining chromatin overnight while rotating at 4 C. The next day blocked Protein A and Protein G beads were added, and rotated for 3 hr. The bound beads were then washed a total of 7 times with chilled wash buffers and immunoprecipitated chromatin was eluted in elution buffer for 30 min at 37 C and cleaned up with ChIP DNA Clean and Concentrator kit (Zymo Research, Irvine, CA). Libraries were prepared for sequencing using the NuGEN ultralow library prep kit (NuGEN, San Carlos, CA) following manufacturer's instructions and sequenced on an Illumina HiSeq 2500 at Hudson Alpha to collect 10.3–10.5 and 12.2–14.5 paired-end 50 bp reads for pull-down and input samples respectively.

Raw reads were trimmed to remove adapter sequences and reads with fewer than 18 bp after adapter trimming using the program cutadapt. These trimmed reads were then mapped to the *X. maculatus* reference genome with bowtie2 (*Langmead and Salzberg, 2012*) and the resulting bam file was sorted with samtools (*Li et al., 2009*). Homer (*Heinz et al., 2010*) was used to generate bigWig files and call peaks using the option style –factor. We also performed the analysis using the option style –histone and found that the results were qualitatively similar. Peak files were converted to bed files and bedtools2 (*Quinlan and Hall, 2010*) was used to analyze overlap between the locations of H3K4me3 peaks and predicted PRDM9 binding motifs in the swordtail genome (see above). Based on Homer analysis, which identified 20,662 peaks in the testis and 15,050 in the liver, the IP efficiency was estimated to be 38% for the testis sample and 40% for the liver sample; the peak width was estimated to be 229 bp for the testis sample and 238 for the liver sample.

Having identified H3K4me3 peaks in testis and liver tissue, we next asked about the relationship between these peaks and predicted PRDM9 binding sites (see above). If PRDM9 is generating H3K4me3 peaks during meiosis, we expect to see an association between predicted PRDM9 binding motifs in the swordtail genome and H3K4me3 peaks. To test for such an association, we generated 500 null motifs by randomly shuffling without replacement the position weight matrix of the *X. birchmanni* PRDM9 and re-running FIMO as described above. We then asked how frequently randomly shuffled PRDM9 motifs overlap H3K4me3 peaks compared to the real motif. We found that no

evidence that the real motif overlapped H3K4me3 peaks more frequently than the shuffled versions of the motif (*Figure 4C*).

As a secondary approach, we compared H3K4me3 peaks that are specific to the testis to H3K4me3 peaks that are specific to the liver, defined as peaks in the testis where there is no over-lapping peak in the liver. Using a Chi-squared test, we asked whether H3K4me3 peaks found only in the testis are more likely to overlap a PRDM9 binding motif than those that are liver specific (where the definition is analogous) (*Figure 4*). Because the size of H3K4me3 peaks will impact the expected overlap with PRDM9 binding motifs, we also constrained the size of the H3K4me3 peaks in the liver analysis to be the same as that inferred from the testis using the –size flag in homer (229 bp). Results were not qualitatively different with the original analysis, using liver H3K4me3 peaks that were inferred to be 238 bp. Counterintuitively liver-specific H3K4me3 peaks appear to overlap predicted PRDM9 motifs more often than testes-specific peaks ($\chi$ = 14.8; p=1.2e-4). However, performing this same analysis with the 500 null motifs (generated as described above), we found that liver-specific peaks were significantly enriched in shuffled motifs in 85% of simulations (at the 0.05 level). This analysis suggests that base composition differences between liver and testes-specific H3K4me3 peaks explain the difference in overlap results.

We also repeated the above analysis for clusters of three ZFs in the swordtail PRDM9 ZF array, using a smaller number of shuffled sequences (n = 20). We observed the same qualitative patterns for each of the ZF clusters as reported above.

Finally, we used a third approach to ask about the association of H3K4me3 peaks and PRDM9 binding sites. We generated five replicate datasets of H3K4me3 sequences and their flanking 250 bp regions from both the testis and the liver. We ran the program MEME to predict motifs enriched in the testis-specific H3K4me3 peaks using the liver as a background sequence set on these five rep-licate datasets. We then examined the top ten predicted motifs to ask whether any of these motifs resembled the computationally predicted PRDM9 binding motifs (*Figure 4—figure supplement 5*).

The above analyses suggest that in swordtail fish, PRDM9 does not make H3K4me3 marks but they do not indicate whether H3K4me3 peaks are associated with recombination events in sword-tails. We therefore verified that recombination rates in 10 kb windows are significantly correlated with the distance of that window to the nearest H3K4me3 peak (rho = −0.072, p=2.3e-69; *Figure 4*). This relationship weakens but remains significant when accounting for distance both to TSSs and CGIs by a partial correlation analysis (rho = −0.026, p=5.4e-10). Furthermore, windows that contain a testis-specific H3K4me3 peak have a higher recombination rate than windows that contain a liver-specific peak (*Figure 4—figure supplement 4*). Finally, there is a significant positive correlation between the number of bp in a 10 kb window overlapping an H3K4me3 peak and the number of recombination events observed in that window in the testis but not in the liver (testis: rho = 0.044, p=2.8e-29; liver: rho = 0.002, p=0.66). Together, these analyses suggest that a relationship between H3K4me3 peaks and recombination exists in swordtails, but not one mediated through PRDM9 binding.

## Acknowledgements

We thank the federal government of Mexico for permission to collect fish under a scientific collecting permit to Guillermina Alcaraz (PPF/DGOPA-173/14). We are grateful to Dana Pe'er for generous use of lab space, Joe Derisi for sending us python tissue, Ammon Corl and Rasmus Nielsen for access to additional lizard transcriptomes, and Nick Altemose, Peter Donnelly, Scott Keeney, Simon Myers, Laure Segurel, Guy Sella, Sonal Singhal and members of the Pickrell, Przeworski and Sella labs for helpful discussions. This project was supported by R01 GM83098 grant to MP and NSF DDIG DEB-1405232 to MS.

## Additional information

### Competing interests

MP: Reviewing editor, *eLife*. The other authors declare that no competing interests exist.

## Funding

| Funder | Grant reference number | Author |
|---|---|---|
| National Institutes of Health | R01 GM83098 | Molly Przeworski |
| National Science Foundation | DDIG DEB-1405232 | Molly Schumer |

The funders had no role in study design, data collection and interpretation, or the decision to submit the work for publication.

## Author contributions

ZB, Conceptualization, Data curation, Visualization, Methodology, Writing—original draft, Writing—review and editing; MS, Conceptualization, Data curation, Funding acquisition, Visualization, Methodology, Writing—original draft, Writing—review and editing; YH, Data curation, Methodology, Writing—review and editing; LB, CH, Data curation, Methodology; GGR, Resources, Supervision; MP, Conceptualization, Resources, Supervision, Funding acquisition, Project administration, Writing—review and editing

## Author ORCIDs

Zachary Baker, http://orcid.org/0000-0002-1540-0731
Molly Schumer, http://orcid.org/0000-0002-2075-5668
Molly Przeworski, http://orcid.org/0000-0002-5369-9009

## Ethics

Animal experimentation: Animals used for this study were handled according to the approved institutional animal care and use committee (IACUC) protocol AUP# - IACUC 2013-0168 of Texas A&M University. All individuals used for dissections were first treated with MS-222 for anesthesia to minimize suffering before being euthanized.

## Additional files

### Supplementary files

• Supplementary file 1. (A) PRDM9 orthologs identified in RefSeq and whole genome databases. Includes which amino acids align to each of three catalytic tyrosine residues of the human PRDM9 SET domain for each PRDM9 ortholog. (B) Genomes targeted for the PRDM9 search. Major groups or individual species lacking PRDM9 in RefSeq were targeted for further analysis of their whole genome sequences, with the exception of previously reported bird and crocodilian losses. Species included and results of this search are reported here.

• Supplementary file 2. (A) Accession numbers and assembly descriptions of publicly available testes RNAseq samples used for de novo assembly and assessment of PRDM9 expression. N50 describes the shortest contig length in which 50% of the assembled transcriptome is contained. (B) Summary of expression results of PRDM9 in the testis in representative species from major taxa. Only species that passed the core recombination protein quality test (see Materials and methods, *Supplementary file 2D*) are included in this table, with the exception of cases, indicated with asterisks, in which PRDM9 was detected but one or more conserved recombination proteins were not. (C) Results of a *rpsblast* search of assembled transcriptomes and a reciprocal best *blast* test to PRMD9. Domain structures found in transcripts that blasted to PRDM9 for each species are also listed. (D) Results of the core recombination protein test for each species for which a transcriptome was assembled. Blue shading indicates that a reciprocal best *blast* test did not identify the gene in the transcriptome.

• Supplementary file 3. (A) Rates of amino acid evolution in SET domains of representative PRDM9 orthologs lacking other functional domains. To determine whether PRDM9 orthologs lacking functional domains are non-functional, we compared rates of evolution between each PRDM9 ortholog missing a domain and another sequence (listed here) with the complete domain structure. The number of aligned bases and the results of a likelihood ratio test of non-neutral versus neutral evolution

are also shown. See Methods for details. (B) Amino acid diversity levels of PRDM9 ZF arrays and the proportion localized to known DNA-binding residues. Columns labeled V1-V28 indicate the amount of amino acid diversity observed at each amino acid in the ZF array. For each gene, we also report the ranking of this proportion relative to all other C2H2 ZF genes from the same species, when such a ranking was feasible. This table additionally includes the average percent DNA identity between ZFs used in our analysis of rapid evolution. (C) Results of the likelihood ratio test of neutral versus not non-neutral evolution along the SET domain of mammalian PRDM9 orthologs lacking a KRAB or SSXRD domain, as annotated in RefSeq (see Materials and methods). We also indicate whether another annotated ortholog exists with a KRAB domain.

• Supplementary file 4. R script to convert GenPept/GenBank files for RefSeq genes into table format.

• Supplementary file 5. Shell script to perform reciprocal best blast search of transcripts from de novo assembly of testis transcriptomes.

### Major datasets

The following datasets were generated:

| Author(s) | Year | Dataset title | Dataset URL | Database, license, and accessibility information |
|---|---|---|---|---|
| Baker Z, Schumer M, Haba Y, Holland C, Rosenthal GG, Przeworski M | 2016 | Xiphophorus birchmanni and Xiphophorus malinche RNAseq data Raw sequence reads | https://www.ncbi.nlm.nih.gov/bioproject/PRJNA358086 | Publicly available at the NCBI BioProject (accession no: PRJNA358086) |
| Baker Z, Schumer M, Haba Y, Bashkirova L, Holland C, Rosenthal GG, Przeworski M | 2017 | Domain Alignments from: Repeated losses of PRDM9-directed recombination despite the conservation of PRDM9 across vertebrates | http://dx.doi.org/10.5061/dryad.gf7r3 | Available at Dryad Digital Repository under a CC0 Public Domain Dedication |

The following previously published datasets were used:

| Author(s) | Year | Dataset title | Dataset URL | Database, license, and accessibility information |
|---|---|---|---|---|
| Song B, Cheng S, Sun Y, Zhong X, Jin J, Guan R, Murphy RW, Che J, Zhang Y, Liu X | 2015 | Anguidae lizard (Ophisaurus gracilis) genome assembly data | http://dx.doi.org/10.5524/100119 | Available at Dryad Digital Repository under a CC0 Public Domain Dedication |
| Xiong Z, Li F, Li Q, Zhou L, Gamble T, Zheng J, Kui L, Li C, Li S, Yang H, Zhang G | 2016 | Supporting data for | http://dx.doi.org/10.5524/100246 | Available at Dryad Digital Repository under a CC0 Public Domain Dedication |
| Bradnam KR, Fass JN, Alexandrov A, Baranay P, Bechner M, Birol I, Boisvert S, Chapman JA, Chapuis G, Chikhi R, Chitsaz H, Chou WC, Corbeil J, Del Fabbro C, Docking TR, Durbin R, Earl D, Emrich S, Fedotov P, Fonseca NA, Ganapathy G, Gibbs RA, Gnerre S, Godzaridis E, Goldstein S, Haimel | 2013 | Assemblathon 2 assemblies | http://dx.doi.org/10.5524/100060 | Available at Dryad Digital Repository under a CC0 Public Domain Dedication |

| M, Hall G, Haussler D, Hiatt JB, Ho IY, Howard J, Hunt M, Jackman SD, Jaffe DB, Jarvis ED, Jiang H, Kazakov S, Kersey PJ, Kitzman JO, Knight JR, Koren S, Lam TW, Lavenier D, Laviolette F, Li Y, Li Z, Liu B, Liu Y, Luo R, Maccallum I, Macmanes MD, Maillet N, Melnikov S, Naquin D, Ning Z, Otto TD, Paten B, Paulo OS, Phillippy AM, Pina-Martins F, Place M, Przybylski D, Qin X, Qu C, Ribeiro FJ, Richards S, Rokhsar DS, Ruby JG, Scalabrin S, Schatz MC, Schwartz DC, Sergushichev A, Sharpe T, Shaw TI, Shendure J, Shi Y, Simpson JT, Song H, Tsarev F, Vezzi F, Vicedomini R, Vieira BM, Wang J, Worley KC, Yin S, Yiu SM, Yuan J, Zhang G, Zhang H, Zhou S, Korf IF | | | | |
| Georges A, Li Q, Lian J, O'Meally D, Deakin J, Wang Z, Zhang P, Fujita M, Patel HR, Holleley CE, Zhou Y, Zhang X, Matsubara K, Waters P, Graves JA, Sarre SD, Zhang G | 2015 | The genome of the Australian dragon lizard Pogona vitticeps | http://dx.doi.org/10.5524/100166 | Available at Dryad Digital Repository under a CC0 Public Domain Dedication |

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
