## [Decision Letter]

Thank you for submitting your article "Repeated losses of PRDM9-directed recombination despite the conservation of PRDM9 across vertebrates" for consideration by *eLife*. Your article has been favorably evaluated by Patricia Wittkopp (Senior Editor) and three reviewers, one of whom, Bernard de Massy (Reviewer #1), is a member of our Board of Reviewing Editors.

The reviewers have discussed the reviews with one another and the Reviewing Editor has drafted this summary of what we consider to be essential results that could allow us to recommend your work for publication in *eLife*. We ask that you respond to the essential revisions discussed below and provide an action plan and an estimate of the time it would take to complete these tasks. We will share your response with the editor and reviewers who will provide advice to guide your efforts.

Summary:

In this manuscript, the authors perform extensive computational analyses to recover 227 PRDM9 orthologs from 149 species. They confirm that PRDM9 has been lost independently in several lineages. They show that the domain architecture of PRDM9 in placental mammals (KRAB, SSXRD, PR/SET and ZF) corresponds to the ancestral state, at the root of vertebrates. They also identified many lineages in which PRDM9 orthologs have lost the KRAB and/or SSXRD domains. These 'truncated' PRDM9 orthologs are subject to strong negative selection, which implies that they are functional. Their zinc finger domain does not show however evidence for positive selection. This observation leads to the conclusion that the presence of KRAB/SSXRD domains is most probably required for the DSB-targeting function of PRDM9. To test this prediction, a recombination map of swordtail fish, which has a truncated PRDM9 has been generated, and the authors conclude that this map shows a pattern similar to that of Prdm9 knock-outs.

Essential revisions:

The phylogeny of Prdm9 is very convincing and the reviewers recognize the quality of this work which provides important information about maintenance or loss of Prdm9 in vertebrates. The association between the presence of KRAB/SSXRD domains and the rate of evolution of the ZF domain is clear, and the authors propose that the KRAB and SSXRD domains are required for the DSB-targeting function of PRDM9. However the conclusion drawn by the authors that recombination sites are preferentially located near TSS in swordtail fish similarly to other species lacking Prdm9 is not convincing; First the increase of recombination rate near TSS is very weak (about 1.2x) compared to that observed in dog (about 4-7x, see Figure 2 and Figure 4 in Auton et al. 2013 Plos Genet) or finches (about 2-3x, see Figure 5 in Singhal et al. 2015 Science). Moreover, the increase in recombination rate in the vicinity of TSS and CGIs in swordtail fish is similar to that observed in human and chimpanzee (see Figure 3 in Auton et al. 2012 Science), two species that have PRDM9-directed hotspots. In addition, the power of detecting an increase of recombination near motifs is limited by the resolution of their map. Having a zinc finger with a normal evolution rate does not automatically means that PRDM9 is not involved in targeting recombination. Other mechanisms may be used such as DNA binding at distance from DSB sites for instance and one cannot interpret the data with only two possible hypotheses to be tested: “the motif mode” or “the promoter/CpG mode”. It is thus essential to obtain accurate mapping data if one wishes to answer this question.

The editor and reviewers feel you may already have the data in hand (with more sequencing data from your fish hybrid and with sequencing/poylmorphism data on other species if available) in which case you could potentially answer these major concerns, or have the possibility to rapidly obtain a high resolution genetic map for the swordtail fish (or another species of interest).

In conclusion, the phylogeny is of great interest but by itself is better suited for a more specialized journal. A major revision with clear conclusions on the functional consequences on the localization of recombination sites in species such as swordtail fish expressing a truncated Prdm9 and possibly in a non-mammalian species expressing a full length Prdm9 would be needed for publication in *eLife*. The authors may want to consider whether this request is compatible with a revision time, which is normally recommended not to exceed two months.

---

## [Author Response]

*Essential revisions:*

*The phylogeny of Prdm9 is very convincing and the reviewers recognize the quality of this work which provides important information about maintenance or loss of Prdm9 in vertebrates. The association between the presence of KRAB/SSXRD domains and the rate of evolution of the ZF domain is clear, and the authors propose that the KRAB and SSXRD domains are required for the DSB-targeting function of PRDM9. However the conclusion drawn by the authors that recombination sites are preferentially located near TSS in swordtail fish similarly to other species lacking Prdm9 is not convincing; First the increase of recombination rate near TSS is very weak (about 1.2x) compared to that observed in dog (about 4-7x, see Figure 2 and Figure 4 in Auton et al. 2013 Plos Genet) or finches (about 2-3x, see Figure 5 in Singhal et al. 2015 Science). Moreover, the increase in recombination rate in the vicinity of TSS and CGIs in swordtail fish is similar to that observed in human and chimpanzee (see Figure 3 in Auton et al. 2012 Science), two species that have PRDM9-directed hotspots. In addition, the power of detecting an increase of recombination near motifs is limited by the resolution of their map. Having a zinc finger with a normal evolution rate does not automatically means that PRDM9 is not involved in targeting recombination. Other mechanisms may be used such as DNA binding at distance from DSB sites for instance and one cannot interpret the data with only two possible hypotheses to be tested: “the motif mode” or “the promoter/CpG mode”. It is thus essential to obtain accurate mapping data if one wishes to answer this question.*

*The editor and reviewers feel you may already have the data in hand (with more sequencing data from your fish hybrid and with sequencing/poylmorphism data on other species if available) in which case you could potentially answer these major concerns, or have the possibility to rapidly obtain a high resolution genetic map for the swordtail fish (or another species of interest).*

*In conclusion, the phylogeny is of great interest but by itself is better suited for a more specialized journal. A major revision with clear conclusions on the functional consequences on the localization of recombination sites in species such as swordtail fish expressing a truncated Prdm9 and possibly in a non-mammalian species expressing a full length Prdm9 would be needed for publication in eLife. The authors may want to consider whether this request is compatible with a revision time, which is normally recommended not to exceed two months.*

In response to these concerns, we have added a number of analyses. Notably:

We collected additional genomic data in the hybrid swordtail population (specifically ~1X genome-‐wide coverage for 268 individuals) and have thereby increased the resolution of our genetic map (resolving crossover events to a median of 13 kb). The increase near the TSS is higher than what we initially reported, but weaker than in some other species without PRDM9 (see Figure 4 and Figure 5). We believe the smaller peak in swordtail fish reflects the lower resolution of our map, consistent with it being higher now than what it was previously (and with differences between pedigree and LD map

in dogs, for example). In that sense, maps from different species are not strictly comparable, a caveat we now state explicitly in the text. We note that another complication in interpreting the depth of troughs and peaks is that maps based on LD confound differences in Ne along the genome with variation in recombination rates; nonetheless, we show our findings to be robust to the use of pedigree/admixture maps or LD-‐based maps (Figure 5—figure supplement 1).

These caveats notwithstanding, we have conducted a comparative analysis of recombination maps in four species with an intact PRDM9 (human, gorilla, mouse, sheep) and three species with no PRDM9 (zebrafinch, long-‐tailed finch, dog) to contrast those patterns with the ones seen in swordtail fish. This analysis shows that swordtail fish behave like PRDM9 knockouts with regard to patterns of recombination around the TSS and CpG islands (Figure 5), and highlights qualitative differences between vertebrates with and without PRDM9 among species studied to date. We agree that this observation does not rule out other roles of PRDM9 in recombination (i.e., roles unlike the ones it plays in mammals).

Based on findings from yeast to mice, we would expect that a role for PRDM9 in recombination would be associated with H3K4me3 marks. For that reason, we have added H3K4me3 chip-‐seq experiments from swordtail testis and liver tissue samples. Consistent with our expectation, we see increased recombination near H3K4me3 peaks in testes, and no effect of the TSS once these peaks and the locations of CpG islands are taken into account (Figure 4, Figure 4—figure supplement 2).

Intriguingly, however, when we analyzed the conservation of catalytic residues of the SET domain for each PRDM9 ortholog in our dataset, we found that ‘PRDM9β’ genes such as the one carried by swordtail fish lack conservation of key residues (see Figure 3).

Moreover, analyzing the correspondence of H3K4me3 peaks and the predicted PRDM9 binding motif in swordtails, we find no association between these H3K4me3 peaks and the predicted PRDM9 binding motif (Figure 4—figure supplement 5). Together 4) and 5) suggest that the SET domain does not lay down H3K4me3 marks in swordtail fish.

Together, we believed these different lines of evidence provide compelling evidence that swordtail fish, although they carry a constrained, truncated ortholog to PRDM9, do not use PRDM9 to direct recombination, at least through any mechanism that has been reported.